# Impact of procedural variability and study design quality on the efficacy of cell-based therapies for heart failure - a meta-analysis

Zhiyi Xu[1], Sebastian Neuber[1,2,3], Timo Nazari-Shafti[1,2,3,4], Zihou Liu[1], Fengquan Dong[5], Christof Stamm[1,2,3,6] *

1 Berlin Institute of Health Center for Regenerative Therapies, Charité–Universitätsmedizin Berlin, Berlin, Germany, 2 Department of Cardiothoracic and Vascular Surgery, German Heart Center Berlin, Berlin, Germany, 3 German Centre for Cardiovascular Research, Partner Site Berlin, Berlin, Germany, 4 Berlin Institute of Health at Charité, Universitätsmedizin Berlin, Berlin, Germany, 5 Department of Cardiology, Shenzhen University General Hospital, Shenzhen, Guangdong, China, 6 Helmholtz Zentrum Geesthacht, Institut für Aktive Polymere, Teltow, Germany

* stamm@dhzb.de

## Abstract

### Background

Cell-based therapy has long been considered a promising strategy for the treatment of heart failure (HF). However, its effectiveness in the clinical setting is now doubted. Because previous meta-analyses provided conflicting results, we sought to review all available data focusing on cell type and trial design.

### Methods and findings

The electronic databases PubMed, Cochrane library, ClinicalTrials.gov, and EudraCT were searched for randomized controlled trials (RCTs) utilizing cell therapy for HF patients from January 1, 2000 to December 31, 2020. Forty-three RCTs with 2855 participants were identified. The quality of the reported study design was assessed by evaluating the risk-of-bias (ROB). Primary outcomes were defined as mortality rate and left ventricular ejection fraction (LVEF) change from baseline. Secondary outcomes included both heart function data and clinical symptoms/events. Between-study heterogeneity was assessed using the I2 index. Subgroup analysis was performed based on HF type, cell source, cell origin, cell type, cell processing, type of surgical intervention, cell delivery routes, cell dose, and follow-up duration. Only 10 of the 43 studies had a low ROB for all method- and outcome parameters. A higher ROB was associated with a greater increase in LVEF. Overall, there was no impact on mortality for up to 12 months follow-up, and a clinically irrelevant average LVEF increase by LVEF (2.4%, 95% CI = 0.75−4.05, p = 0.004). Freshly isolated, primary cells tended to produce better outcomes than cultured cell products, but there was no clear impact of the cell source tissue, bone marrow cell phenotype or cell chricdose (raw or normalized for CD34+ cells). A meaningful increase in LVEF was only observed when cell therapy was combined with myocardial revascularization.

**Data Availability Statement:** All relevant data are within the paper and its Supporting Information files.

**Funding:** The authors received no specific funding for this work.

**Competing interests:** The authors have declared that no competing interests exist.

**Abbreviations:** ALDH, aldehyde dehydrogenase; AT, adipose tissue; BM, bone marrow; BMMNC, bone marrow-derived mononuclear cell; BMMSC, bone marrow-derived mesenchymal stromal cell; BNP, brain natriuretic peptide; CABG, coronary artery bypass graft; CDC, cardiac-derived stem cell; CHF, congestive heart failure; CI, confidence interval; CPB, cardiopulmonary bypass; FBS, fetal bovine serum; FUP, follow-up; G-CSF, granulocyte-colony stimulating factor; HF, heart failure; HF-MACE, heart failure-relate d major adverse cardiovascular event; ICI, intracoronary injection; IHD, ischemic heart disease; IMI, intramyocardial injection; LV, left ventricular; LVAD, left ventricular assist device; LVEDD, left ventricular end-diastolic diameter; LVEDV, left ventricular end-diastolic volume; LVEDVI, left ventricular end-diastolic volume index; LVEF, left ventricular ejection fraction; LVESV, left ventricular end-systolic volume; MACE, major adverse cardiovascular event; MPC, myocardial progenitor cell; NT-proBNP, N-terminal pro brain natriuretic peptide; NYHA, New York Heart Association; RCT, randomized controlled trial; RR, risk ratio; SAE, serious adverse event; Std.MD, standard mean difference; TESI, transendocardial stem cell injection; UCMSC, umbilical cord-derived mesenchymal stromal cells.

## Conclusions

The published results suggest a small increase in LVEF following cell therapy for heart failure, but publication bias and methodologic shortcomings need to be taken into account. Given that cardiac cell therapy has now been pursued for 20 years without real progress, further efforts should not be made.

## Study registry number

This meta-analysis is registered at the international prospective register of systematic reviews, number CRD42019118872.

## Introduction

Heart failure (HF) is one of the most common causes of morbidity and mortality worldwide and its prevalence continues to increase [1]. The loss of cardiomyocyte quantity and/or function leads to tissue fibrosis, remodeling of left ventricular geometry, progressive contractile dysfunction and ultimately organ failure. Other than treating discernible underlying causes and palliative medication, only mechanical assist devices and/or heart transplantation help extent life in end-stage HF [2], but still more than 50% of HF patients die within 4 years after diagnosis.

For approximately two decades, stem cell transplantation has been deemed a promising tool for restoring myocardial function in HF [3], overcoming the postnatal cardiomyocyte cell cycle re-entry block that dogmatically thwarts endogenous regeneration. Within months following the publication of the first experimental research reports [4–6], bone marrow cells were delivered into hearts of patients with acute myocardial infarction or chronic ischemic heart disease. A multitude of safety and feasibility trials, compassionate use applications and completely unregulated therapy attempts were soon followed by structured clinical efficacy studies. Most of the early controlled trials, including blinded RCTs, were reported to have a positive result in terms of left ventricular function gain and clinical symptoms, which was also reflected in a number of systematic reviews and meta-analyses [7–10]. However, the design of these trials varied considerably, and numerous inconsistencies were identified in systematic analyses [11], in some instances bordering scientific misconduct [12]. It is therefore difficult, if not impossible, to determine an overall level of evidence to the concept of cardiac cell therapy [13]. Taking the available clinical evidence together, cell-based therapy is relatively safe, with few treatment-related adverse events, and no increase in total serious adverse events compared to placebo [14, 15]. Regarding efficacy, however, many clinicians and researchers agree today that somatic, non-pluripotent progenitor or stem cells are not able to yield a relevant improvement in heart function. However, such therapies continue to be offered to patients and the "promise" of cardiac cell therapy is still upheld by many [16].

We therefore decided to re-appraise the existing body of literature regarding clinical cardiac cell therapy for HF, focusing on both cell type selection and quality of the study design. The biology and pathophysiology of chronic HF differ from that of acute myocardial infarction, and so do the requirements of the potential novel therapies for each, i.e. limiting reperfusion injury versus actual re-generating myocardium. Hence, we chose to concentrate on trials targeting HF rather than acute MI.

## Methods

### Protocol and registration

This systematic review of meta-analysis was carried out based on preferred reporting items for systematic reviews and meta-analyses (PRISMA) guidelines using the PRISMA checklist (S1 Text). The protocol of this systematic review was registered on the international prospective register of systematic reviews (PROSPERO) with registration number CRD42019118872.

### Search strategy and study selection

Studies were searched on PubMed, Cochrane library, ClinicalTrials.gov, and EudraCT from January 1, 2000 to December 31, 2020. The search strategy included the following terms: "heart failure" or "myocardial ischemia" or "myocardial infarction" or "cardiomyopathy" combined with "stem cell transplantation", "stem cells" or "transplantation" (S2 Text). The article type filter was set to clinical trials and research articles. Additional inclusion and exclusion criteria are listed in S1 Table.

### Data extraction

Data were extracted independently by three reviewers using a standardized data extraction form; disagreements were resolved through discussion. When extracting data from multi-arm studies with different interventions, the number of participants in the control group was divided and equally distributed across separate comparisons to avoid repeated entries. Further details on screening and data extraction are given on the PROSPERO website (protocol CRD42019118872).

### Main outcome measures

Primary outcomes were mortality and LVEF change from baseline. Secondary outcomes included HF-related major adverse cardiovascular events (HF-MACEs), serious adverse events (SAEs), left ventricular end-diastolic diameter (LVEDD), left ventricular end-diastolic volume (LVEDV), left ventricular end-diastolic volume index (LVEDVI), serum levels of brain natriuretic peptide (BNP) and N-terminal pro brain natriuretic peptide (NT-proBNP), six-minute walk (6MW) distance, walking speed, New York Heart Association (NYHA) functional status, and quality of life assessed by the Minnesota Living with Heart Failure Questionnaire. HF-MACEs were defined as a composite of non-fatal stroke, non-fatal myocardial infarction, cardiac death, decompensated HF, and persistent ventricular arrhythmias. Detailed information on primary and secondary outcomes is summarized in S2 Table.

### Estimation of progenitor cell dose

To account for the heterogeneity in cell preparations when evaluating the impact of cell dose on treatment efficacy, we estimated the number of actual progenitor cells in each cell product based on expressions of specific biomarkers. Thus, we were able to compare the presumed "quality" of, for instance, freshly isolated bone marrow (BM) mononuclear cells with a BM-based cell product derived by specific cell selection and in vitro expansion. For skeletal myoblasts, cell numbers were calculated by the expression of CD56 or desmin [17]. For bone marrow-derived mononuclear cells (BMMNCs) and adipose tissue-derived stromal vascular fractions (ATSVFs), CD34 was used for calculation [18, 19]. For the lxmyelocel-T group, a mixture of cell types, including myeloid cells, lymphoid cells, and stromal cells, the number of regenerative cells was calculated based on CD90 expression [20].

## Statistics

Mean difference and 95% confidence intervals (95% CIs) for each reported outcome in the treated versus untreated group were calculated using Stata (version 15) and RevMan (version 5.3). We used a random-effect model and inverse variance weighting for all meta-analyses. For the continuous variables (i.e., LVEF, LVEDV levels), the mean change from baseline was calculated according to the Cochrane handbook [21], de Jong [22], and Hristov [23]. Risk-of-bias (ROB) was evaluated based on the revised Cochrane ROB tool for randomized trials [24]. Potential publication bias was assessed by funnel plot and Egger's test [25]. A p-value of less than 0.05 was considered significant. Between-study heterogeneity was assessed using the $I^2$ index; $I^2 > 50\%$ was considered as substantial heterogeneity. In order to identify sources of heterogeneity, sensitivity analyses and subgroup analyses were performed based on clinical and methodological differences, including cell type, HF type, cell source, cell delivery route, and cell dose. A meta-regression analysis was carried out to evaluate the association between cell dose and primary outcomes.

## Results

### Search results and characteristics of included trials

In total, 31737 records were retrieved from PubMed and Cochrane library and 918 records were retrieved from clinical trial registries (S2 Text). After removing duplicates, 16690 articles were evaluated. Of these, 16638 were subsequently excluded as they failed to meet the predefined inclusion criteria (S1 Table). The inclusion criteria for meta-analysis were met by 43 studies with a total of enrolled 2855 participants [4–6, 26–66]; details can be found in the PRISMA flow diagram (Fig 1). Characteristics of all studies included in this meta-analysis are summarized in Table 1. Subgroups were defined based on HF type, cell source, cell origin, cell type, cell processing, type of surgical intervention, cell delivery routes, delivered cell doses, and follow-up (FUP) visits; details are given in Fig 2.

### Evaluation of ROB and its impact on study outcomes

The risk-of-bias (ROB) was assessed using the Cochrane ROB tool. We found significant differences between the studies in terms of selection, performance, detection, attrition, reporting, and other bias (Fig 3A and 3B). In general, 10 of 43 studies had a low ROB. Adequate random sequence generation and appropriate allocation concealment were done in 30 and 22 clinical trials, respectively. Furthermore, 27 performed adequate blinding of participants and personnel, 36 performed adequate blinding of the assessment outcomes, and 39 and 36 had a low ROB in terms of incomplete outcome data and selective outcome reporting, respectively. We next tested whether a high or unclear ROB affected the results of cell therapy studies. Clinical trials with a high or unclear risk of selection bias showed an LVEF increase by 4% (95% CI = 2.71–5.28) and 3.38% (1.40–5.36) after 6 and 12 months in response to cell treatment, whereas studies with a low risk of selection bias showed no statistically significant LVEF change over time (6 months: 1.53% (-0.35–3.41); 12 months: 1.04% (-1–3.09)) (Fig 3C and 3D). Similar observations were made for studies that were at high or unclear risk of performance bias (Fig 3E and 3F). In contrast, LVEF results were relatively robust to detection bias (Fig 3G and 3H). In sum, due to the high or unclear risk of methodological bias in some studies, it is difficult to evaluate whether cell therapy actually induced a positive LVEF response.

### Overall effects of cell-based treatment

We found no statistically significant beneficial effect of cell therapy in reducing the mortality of HF patients within 6 months (RR = 0.88, 95% CI = 0.59–1.32, p = 0.54, $I^2 = 0\%$) and 12 months

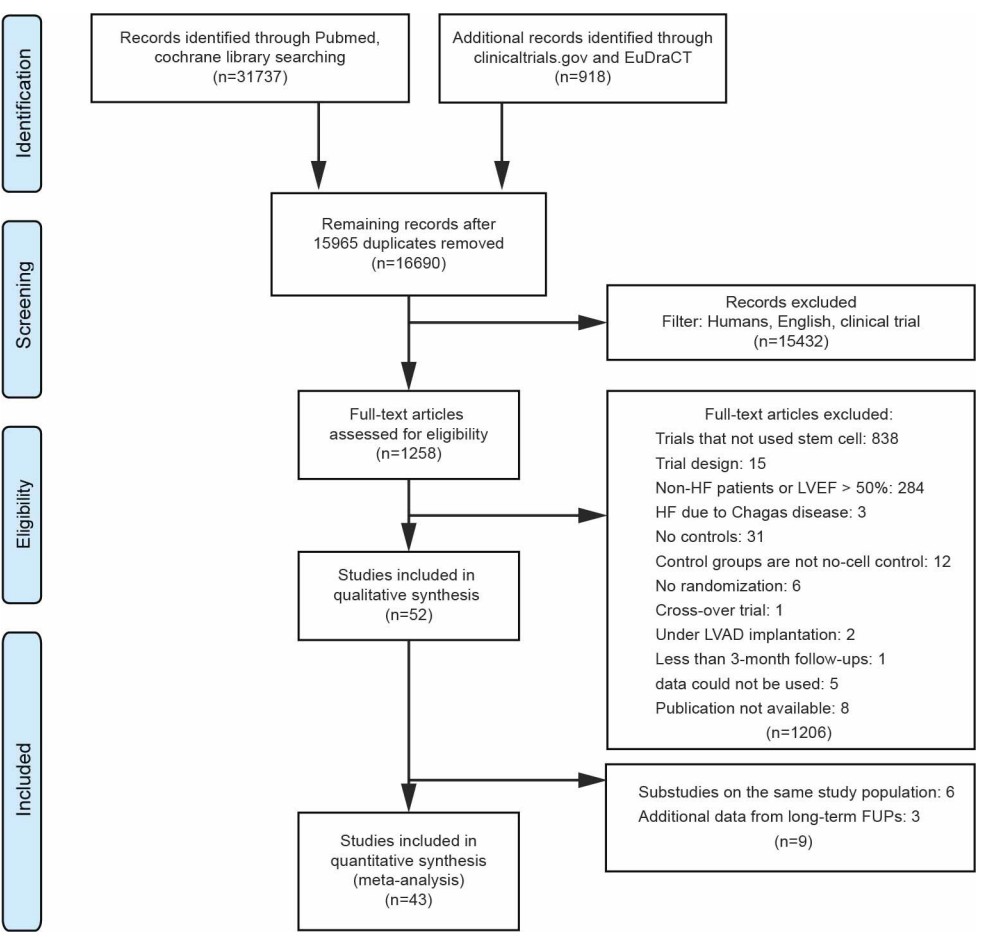

**Fig 1. PRISMA flowchart used in the study selection.**

(RR = 0.74, 95% CI = 0.47−1.15, p = 0.17, $I^2$ = 26%), respectively (Fig 4A). However, at long-term FUP (>12 months, depending on the trial), the mortality rate in patients who received cell therapy was significantly lower (RR = 0.58, 95% CI = 0.41−0.82, p = 0.002, $I^2$ = 9%; Fig 4A). Furthermore, cell therapy had no impact on reducing MACEs (Fig 4B) and SAEs (Fig 4C). Overall, patients treated with cell therapy showed a slightly greater LVEF change compared to controls both after 6 months (2.72%, 95% CI = 1.52−3.93, p<0.001, $I^2$ = 82%; Fig 4D) as well as after 12 months (2.53%, 95% CI = 0.92−4.14, p = 0,002, $I^2$ = 75%; Fig 4D). Given the high inter-trial heterogeneity, which indicates substantial discrepancies between the trials, we used Funnel plots and Egger's tests to evaluate the publication bias. As shown in S1A–S1E Fig, there was no publication bias in the analysis of mortality and short-term LVEF, but there was a significant publication bias in the analysis of LVEF at 12-month FUPs, which is probably due to a significant difference in the proportions of positive and negative results. Patients who received cells had reduced LV end-diastolic dimensions (Fig 4E), eased HF syndromes (Fig 4F), higher exercise tolerance (Fig 4G), lower BNP/NT-proBNP levels (Fig 4H), and a tendency towards lower quality-of-life scores (Fig 4I). Further details on the outcomes after cell therapy are listed in S4 and S5 Tables.

## Impact of cell processing on the outcome of intervention

We next compared the effects of primary cells (28 trials) and *in vitro* cultured cells (15 trials) after transplantation in HF patients. After 6 and 12 months, no reduction in the mortality rate was

Table 1. Characteristics of 43 RCTs included in this meta-analysis.

| Study | Group size | HF type | Cell source | Cell type | Nucleated cell counts (mio) | Cell purity | Progenitor cells (mio) | Ex vivo expansion medium | Intervention | Follow-up visits (months) for safety assessment | Follow-up visits (months) for efficacy assessment |
|---|---|---|---|---|---|---|---|---|---|---|---|
| **Muscle-derived cells** | | | | | | | | | | | |
| **Brickwedel 2013** MAGIC II [65] | 7 | Ischemic | Autologous | Myoblasts | - | - | 400–800 | Cultured but not mentioned | CABG + IMI | 80 | 6, 12, 64–80 |
| **Dib 2009** CAuSMIC [26] | 23 | Ischemic | Autologous | Myoblasts | - | - | 30–600 | 15–20% FBS | TESI | 12 | 0.25, 3, 6, 12 |
| **Duckers 2011** SEISMIC [27] | 47 | Ischemic | Autologous | Myoblasts | 596 ± 194 | CD56$^+$ cells: 57.2 ± 23.9% | >100 | 15–20% FBS | IMI | 6 | 3, 6 |
| **Menasche 2008** MAGIC [36] | 120 | Ischemic | Autologous | Myoblasts | - | 89% | 400–800 | FBS | CABG + IMI | 6 | 6 |
| **Povsic 2011** [49] | 23 | Ischemic | Autologous | Myoblasts | 400–800 | - | 400–800 | FBS | TESI | 6 | 3, 6 |
| **Bone marrow-derived cells** | | | | | | | | | | | |
| **Ang 2008** [46] | 63 | Ischemic | Autologous | BMMNCs | IMI arm: 84 ± 56; ICI arm: 115 ± 73 | - | CD34$^+$ CD117$^+$ cells: IMI arm: 0.14 ± 0.17; ICI arm: 0.25 ± 0.25 | - | CABG + ICI/ IMI | 6 | 6 |
| **Assmus 2013** CELLWAVE [57] | 103 | Ischemic | Autologous | BMMNCs | low-dose BMMNCs 150 ± 77: high-dose BMMNCs: 123 ± 69 | - | 1–10 | - | shockwave + ICI | 4 | 4 |
| **Bartunek 2013** C-CURE [63] | 47 | Ischemic | Autologous | BM-derived cardiopoietic stem cell | 733 | - | 733 | 5% human pooled platelet lysate | TESI | 24 | 6 |
| **Bartunek 2017** CHART-1 [64] | 348 | Ischemic | Autologous | BM-derived cardiopoietic stem cell | 600 | - | up to 480 | 5% human platelet lysate | retention enhanced TESI | 12 | 6.5, 9.75, 12 |
| **Choudhury 2017** REGENERATE-IHD [66] | 70 | Ischemic | Autologous | BMMNCs | 115.1 | - | 1–10 | - | G-CSF + TESI/ ICI | 12 | 6, 12 |
| **Frijak 2018** REMEDIUM [28] | 60 | Non-ischemic | Autologous | BMMNCs | - | - | CD34$^+$ cells: 80 | - | TESI | 6 | 6 |
| **Heldman 2014** TAC-HFT [5] | 65 | Ischemic | Autologous | BMMNCs; BMMSCs | BMMSCs: 200; BMMNCs: 200 | - | *BMMSCs:* >100; *BMMNCs:* 1–10 | 20% FBS | TESI | 12 | 3, 6, 12 |

*(Continued)*

**Table 1.** (Continued)

| Study | Group size | HF type | Cell source | Cell type | Nucleated cell counts (mio) | Cell purity | Progenitor cells (mio) | Ex vivo expansion medium | Intervention | Follow-up visits (months) for safety assessment | Follow-up visits (months) for efficacy assessment |
|---|---|---|---|---|---|---|---|---|---|---|---|
| **Henry 2014** IMPACT-DCM [29] | 61 | Both | Autologous | Lxmyelocel-T | Catheter-DCM: 13 ± 1.6 / IMPACT-DCM: 25 ± 2.6 | CD90+ cells: 5–55% / CD45+CD14+ cells: 45–95% | 1–10 | 20% FBS | IMI/TESI | 24 | 1, 3, 6, 12 |
| **Hu 2011** [31] | 60 | Ischemic | Autologous | BMMNCs | 131.7 ± 106.6 | - | 1–10 | - | CABG + ICI | 6 | 6 |
| **Martino 2015** [33] | 160 | Non-ischemic | Autologous | BMMNCs | 236 ± 271 | CD34+ cells: 2.26% / CD133+ cells: 0.03% | 1–10 | - | ICI | 12 | 6, 12 |
| **Mathiasen 2015** MSC-HF [34] | 60 | Ischemic | Autologous | BMMSCs | 77.5 ± 67.9 | - | 77.5 ± 67.9 | 10% FBS | TESI | 6 | 1, 3, 6 |
| **Maureira 2012** [35] | 14 | Ischemic | Autologous | BMMNCs | 342 ± 41 | CD34+ cells: 3 ± 1% / CD73+ cells: 0.2 ± 0.6% | 10–100 | - | CABG + IMI | 6 | 1, 6 |
| **Nasseri 2014** Cardio133 [37] | 60 | Ischemic | Autologous | BM-derived CD133+ cells | - | CD133+: 73.6% | CD133+ cells: 5.1 | - | CABG + IMI | 34 | 6 |
| **Noiseux 2016** [38] | 33 | Ischemic | Autologous | BM-derived CD133+ cells | - | 63.3 ± 15.5% | CD133+ CD34+ / CD45+ cells: 6.5 ± 3.1 | - | CABG + IMI | 6 | 6 |
| **Patel 2005** [39] | 20 | Ischemic | Autologous | BM-derived CD34+ cells | - | 70% | CD34+ cells: 22 | - | CABG + IMI | up to 6 | 0.25, 1, 3, 6 |
| **Patel 2015** REVIVE [40] | 60 | Both | Autologous | BMMNCs | 3700 ± 900 | - | CD34+ cells: 39.4 | - | ICI | 12 | 1, 3, 6, 12 |
| **Patel 2016** IxCELL-DCM [41] | 126 | Ischemic | Autologous | Lxmyelocel-T | 40–200 | CD45+CD90+ cells: ≥ 90% / CD90+ cells: ≥ 3% | CD90+ cells: 1–10 | 20% FBS | TESI | 12.5 | 3, 6, 12 |
| **Patila 2014** [42] | 39 | Ischemic | Autologous | BMMNCs | 840 | CD34+ cells: 10% / CD34+CD133+ cells: 7.5% | CD34+ cells: 10–100 | - | CABG + IMI | 45.1–72.6 | 12, 45.1–72.6 |
| **Perin 2011** FOCUS-HF [43] | 30 | Ischemic | Autologous | BMMNCs | 484.1 ± 313 | CD34+ cells: 1.5 ±0.4% | 1–10 | - | TESI | 6 | 3, 6 |

(*Continued*)

Table 1. (Continued)

| Study | Group size | HF type | Cell source | Cell type | Nucleated cell counts (mio) | Cell purity | Progenitor cells (mio) | Ex vivo expansion medium | Intervention | Follow-up visits (months) for safety assessment | for efficacy assessment |
|---|---|---|---|---|---|---|---|---|---|---|---|
| **Perin 2012a** FOCUS-Br [44] | 20 | Ischemic | Autologous | Sorted ALDH + / bright BMMNCs | 2.94 ± 1.31 | - | ALDH+ cells: 2.37 ± 1.31 | - | TESI | 6 | 6 |
| **Perin 2012b** FOCUS-CCTRN [4] | 92 | Ischemic | Autologous | BMMNCs | 100 | CD34+ cells: 2.6% / CD133+ cells: 1.2% | 1–10 | - | TESI | 60 | 6 |
| **Perin 2015** [47] | 60 | Both | Allogeneic | STRO3 + MPCs | 25 / 75 / 150 | - | 25 / 75 / 150 | 10% FBS | TESI | 36 | 3, 6, 12 |
| **Pokushalov 2010** ESCAPE [48] | 109 | Ischemic | Autologous | BMMNCs | 41 ± 16 | CD34+ cells: 2.5 ± 1.6% | 1–10 | - | TESI | 12 | 3, 6, 12 |
| **Qi 2018** [50] | 42 | Ischemic | Autologous | BMMNCs | 132.8 ± 94.1 | - | 1–10 | - | ICI | 13 | 12 |
| **Sant'Anna 2014** INTRACELL [51] | 30 | Non-ischemic | Autologous | BMMNCs | 106 ± 43 | CD34+ cells: 1.5 ± 0.7% | 1–10 | - | Transthoracic + IMI | 12 | 3, 6, 9, 12 |
| **Santoso 2014** END-HF [52] | 28 | Ischemic | Autologous | BMMNCs | - | - | CD34+ cells: 2.4 ± 1.2/ml | - | TESI | 36 | 6 |
| **Seth 2006** ABCD [53,54] | 85 | Non-ischemic | Autologous | BMMNCs | 168 ± 96 | - | CD34+ cells: 2.7 ± 1.5 | - | ICI | 36 | 6, 36 |
| **Steinhoff 2017** PERFECT [55] | 82 | Ischemic | Autologous | BM-derived CD133+ cells | - | - | 2.29 ± 1.42 | - | CABG + IMI | 84 | 6 |
| **Trifunovic 2015** [56] | 30 | Ischemic | Autologous | BMMNCs | 70.7 ± 32.4 | - | CD34+ cells: 3.96 ± 2.77 / CD133+ cells: 2.65 ± 1.71 | - | CABG + IMI | 90 | 2, 6, 12, up to 90 |
| **Vrtovec 2011** [6] | 110 | Non-ischemic | Autologous | CD34+ cells | - | - | CD34+ cells: 113 ± 26 | - | G-CSF + ICI | 60 | 3, 6, 9, 12 |
| **Wang 2015** [58] | 90 | Ischemic | Autologous | BMMNCs | 521±44 | - | - | - | CABG + IMI | 6 | 6 |
| **Xiao 2017** [59] | 53 | Non-ischemic | Autologous | BMMNCs / BMMSCs | BMMNCs: 510 ± 200 / BMMSCs: 490 ± 170 | - | BMMNCs: 1–10 / BMMSCs: >100 | BMMSC: 15% autologous serum | ICI | 12 | 3, 12 |
| **Zhao 2008** [60] | 36 | Ischemic | autologous | BMMNCs | 659 ± 512 | - | 10–100 | - | CABG + IMI | 1, 3, 6 | 1, 3, 6 |
| **Adipose tissue-derived stem cells** | | | | | | | | | | | |
| **Henry 2017** ATHENA [30] | 31 | Ischemic | Autologous | AT-SVF | 40 ± 9 | - | - | - | TESI | 60 | 1, 3, 6, 12 |

(Continued)

**Table 1.** (Continued)

| Study | Group size | HF type | Cell source | Cell type | Nucleated cell counts (mio) | Cell purity | Progenitor cells (mio) | Ex vivo expansion medium | Intervention | Follow-up visits (months) | |
|---|---|---|---|---|---|---|---|---|---|---|---|
| | | | | | | | | | | for safety assessment | for efficacy assessment |
| **Perin 2014 PRECISE** [45] | 27 | Ischemic | Autologous | AT-SVF | 42 | CD34+ cells: 70.4% | *10–100* | - | TESI | 36 | 6, 12, 18 |
| **Perinatal cells** | | | | | | | | | | | |
| **Bartolucci 2017 RIMECARD** [62] | 30 | Non-ischemic | Allogeneic | UCMSCs | - | - | 1 per kg | 10% AB plasma | IVI | 12 | 3, 6, 12 |
| **Zhao 2015** [61] | 59 | Non-ischemic | Allogeneic | UCMSCs | - | - | - | 10% FBS | ICI | 6 | 1, 6 |
| **Cardiac-derived cells** | | | | | | | | | | | |
| **Makkar 2020** [32] | 142 | Ischemic | Allogeneic | CDCs | - | - | 25 | 10% FBS or 20% other specimen | ICI | 12 | 6, 12 |

The number of progenitor cells were used to calculate cell dose dependent effects of treatment (Fig 11). In trials using BMMNCs where the CD34+ cell fraction was not reported the fraction of progenitor cells was set at 1% (these numbers are marked by italic font). This estimate was based on data published on BMMNCs characterizations in patients with heart failure[67].

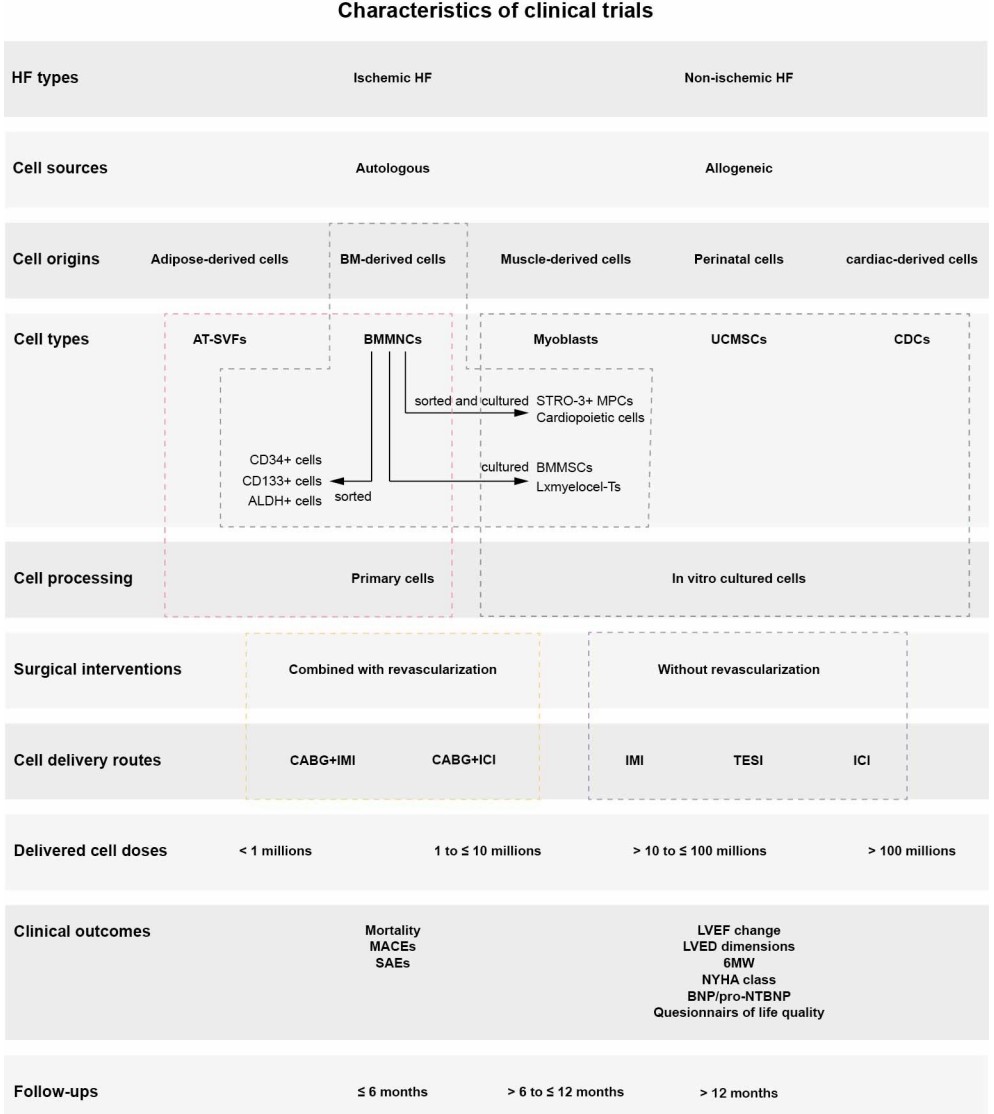

**Fig 2. Subgroups of HF patients.**

found for either cell type (Fig 5A and 5B). However, at long-term FU, the administration of primary cells resulted in a lower mortality rate (RR = 0.59, 95% CI = 0.38–0.91, p = 0.02; Fig 5C). In addition, both primary and *in vitro* cultured cells improved myocardial contractility after 6 months (Fig 5D), but only primary cells resulted in an LVEF improvement after 12 months (3.11%, 95% CI = 1.04–5.18, p = 0.003; Fig 5E). Furthermore, LVED dimension was not different within 6 months after transplantation of either primary or in vitro cultured cells (Fig 5F), but was significantly reduced after 12 months in patients treated with primary cells (Fig 5G). Patients who received cells cultured *in vitro* showed a slightly reduced NYHA score after 6 months (Fig 5H), but no change after 12 months (Fig 5I). Further details are given in S4 and S5 Tables.

## Impact of cell origin on the outcome of intervention

The 43 RCTs included in this meta-analysis were grouped based on cell origin (Fig 2 and Table 1). In five studies skeletal muscle-derived cells were used (so-called myoblasts; 220

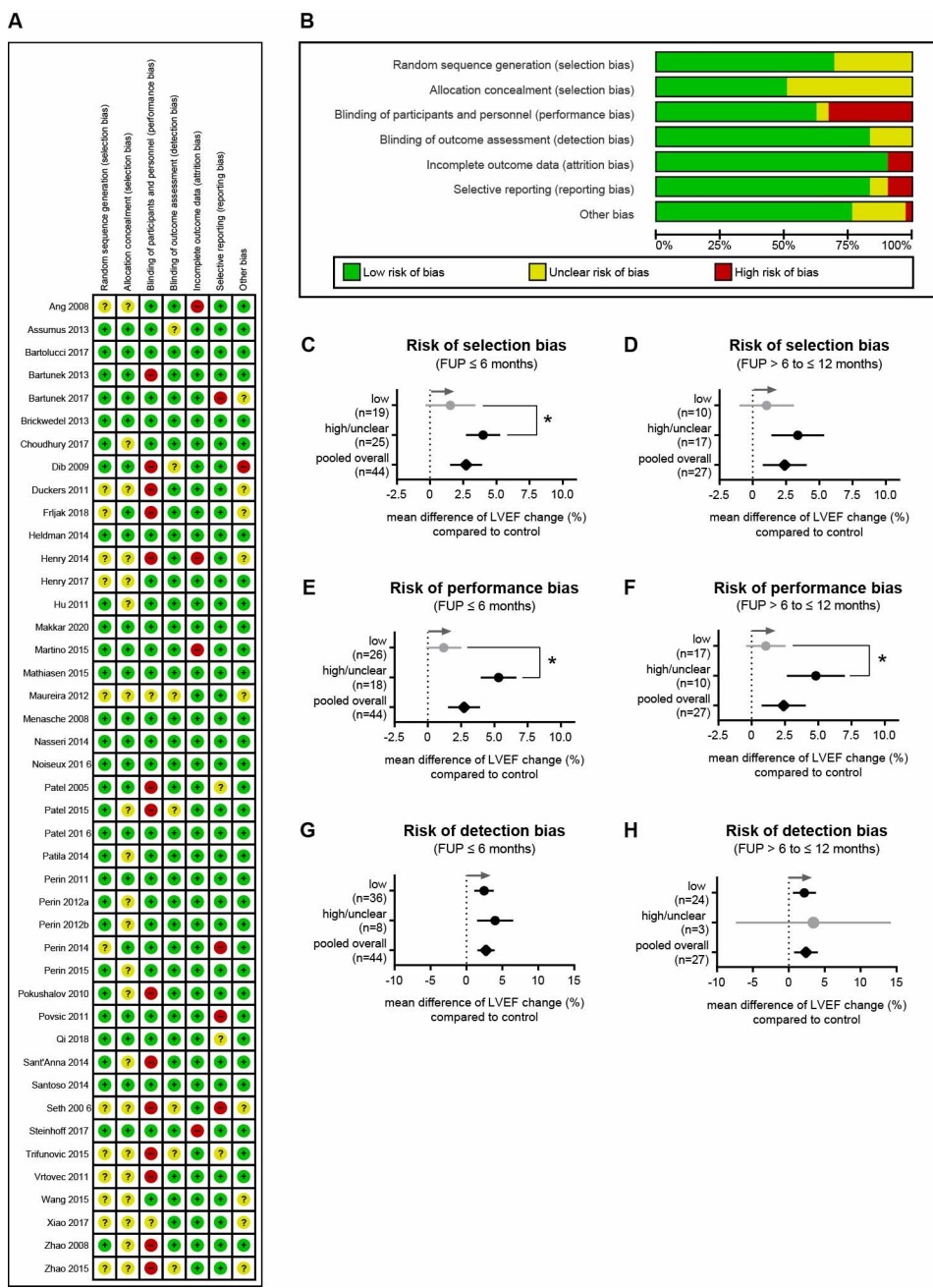

**Fig 3. ROBs in included studies and their impact on study outcomes.** (A) Authors' judgments about each ROB item for each included study. Green, red, and yellow dots, respectively, indicate low ROB, high ROB, and unclear ROB. (B) Judgments about each ROB item presented as percentages across all 43 included studies. (C, D) Risk of selection bias on the LVEF change in response to cell therapy after 6 (p = 0.03) and 12 months compared to untreated controls. (E, F) Risk of performance bias on the LVEF change in response to cell therapy after 6 (p<0.001) and 12 (p = 0.005) months compared to untreated controls. (G, H) Risk of detection bias on the LVEF change in response to cell therapy after 6 and 12 months compared to untreated controls. Stars indicate significant changes, gray arrows represent the trend in favor of cell therapy, and n indicates the number of studies considered. Information on which studies were included in the analysis is shown in S3 Table.

participants), 33 studies evaluated bone marrow-derived cells (2346 participants), two studies assessed adipose tissue-derived cells (58 participants), two studies analyzed perinatal stem cells

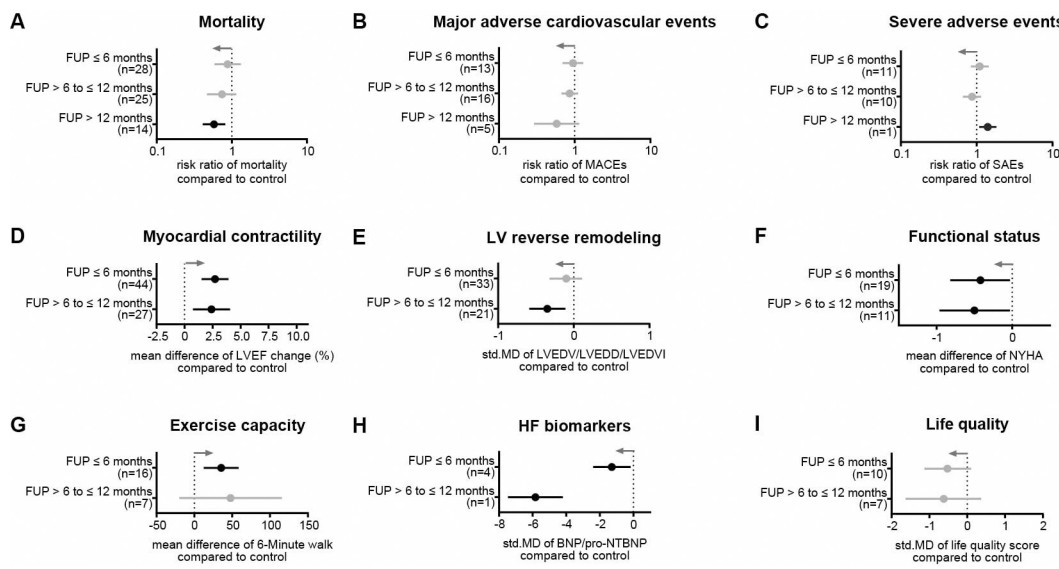

**Fig 4. Overall results of clinical safety and efficacy outcomes after cell therapy.** Given parameters are (A) mortality, (B) MACEs, (C) SAEs, (D) LVEF change from baseline, (E) LV end-diastolic dimensions, (F) NYHA functional class, (G) exercise capacity, (H) HF biomarkers BNP/NT-proBNP, and (I) scores of life quality. Gray arrows represent the trend in favor of cell therapy and n indicates the number of studies considered. Information on which studies were included in the analysis is shown in S3 Table.

(89 participants), and one study investigated cardiac-derived cells (142 participants). When analyzing muscle-derived, BM-derived and perinatal cells, there was no difference in mortality between cell therapy and control groups at 6 months (Fig 6A). Similarly, BM-derived, AT-derived and perinatal cells showed no effect on reducing death rates within 12 months (Fig 6B). At long-term FU, however, the administration of BM-derived stem cells correlated with a reduced mortality rate (RR = 0.58, 95% CI = 0.41−0.82, p = 0.004; Fig 6C). Of note, only BM-derived and perinatal stem cells appeared to improve myocardial contractility after short- and long-term FUP visits (Fig 6D and 6E). In addition, only cardiac-derived cells showed a difference in LVED dimensions after 6 months compared to controls (Fig 6F) and only BM-derived cells had the same impact after 12 months (Fig 6G). For functional status, no cells showed significant improvements after 6 months (Fig 6H) and only muscle-derived cells showed a difference in NYHA score compared to controls after 12 months (Fig 6I). Furthermore, BNP/NT-proBNP levels, exercise capacity, and incidence of MACEs were positively influenced by BM-derived cells (S4 and S5 Tables). In contrast, transplantation of muscle-, AT-, and cardiac-derived stem cells resulted in almost no benefits in terms of primary outcomes. Therefore, these cell types were not included in further analyses. Perinatal cell therapy trials were excluded from further analyses because the statistical power was poor due to very limited data.

## Impact of BM cell phenotype on the outcome of intervention

In the 33 trials using BM-derived cells, very different cell products were used. We therefore analyzed the effect of BM mononuclear cells (BMMNC, 19 studies and 2 extra arms; 1259 participants), CD34-positive cells (2 studies, 130 participants), CD133-positive cells (3 studies, 175 participants), aldehyde dehydrogenase (ALDH)-positive cells (1 study, 20 participants), BMMSCs (1 study and 2 extra arms, 120 participants), cardiopoietic cells (2 studies, 395 participants), STRO-3-positive myocardial progenitor cells (MPCs) (1 study, 60 participants), and lxmyelocel-Ts (2 studies, 187 participants). For none of these cell types, we found a difference

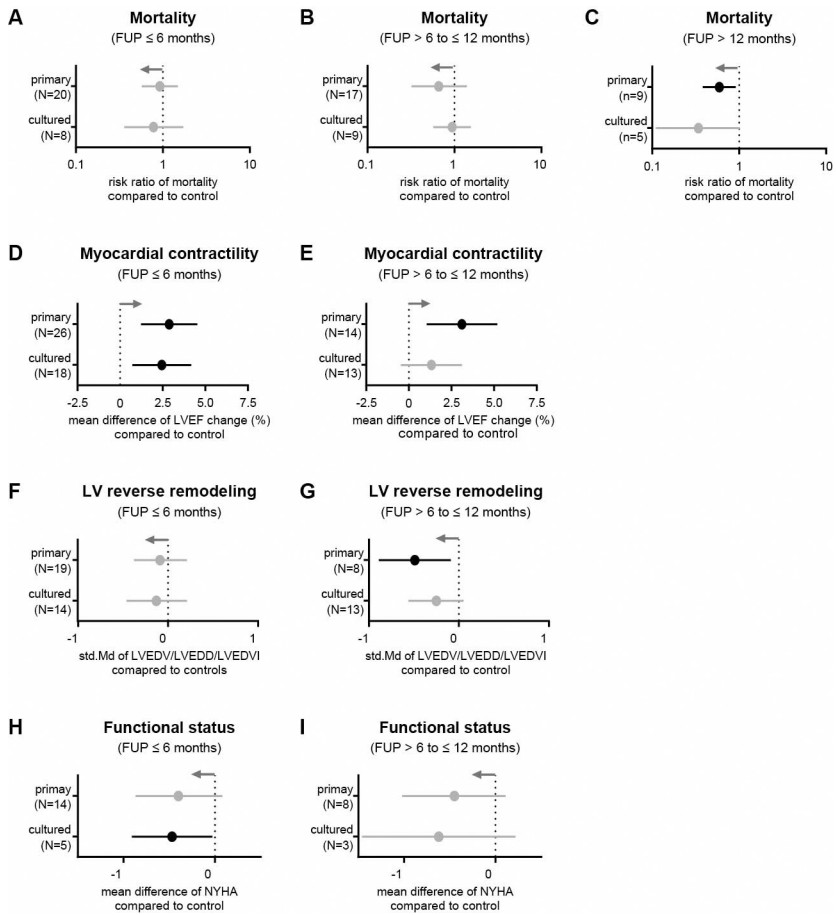

**Fig 5. Comparison of transplanted primary and in vitro cultured cells on clinical outcomes in HF patients.** (A-C) Mortality rates at 6, 12, and more than 12 months. (D, E) Change in LVEF at 6- and 12-month follow-up visits. (F, G) Change in LV end-diastolic dimension at 6- and 12-month follow-up visits. (H, I) NYHA functional status class at 6- and 12-month follow-up visits. Gray arrows represent the trend in favor of cell therapy and n indicates the number of studies considered. Information on which studies were included in the analysis is shown in S3 Table.

in the risk of mortality after 6 or 12 months after transplantation compared to controls (Fig 7A and 7B). At long-term FU, the risk of mortality appeared to be reduced after transplantation of CD34-positive cells (Fig 7C). Following BMMNC treatment patients had an 3.56% increased LVEF at 6 months (95% Cl = 2.24–4.87, p<0.001, Fig 7D) and 2.55% at 12 months (95% Cl = 0.09–5.02, p = 0.04, Fig 7E). Similarly, enriched CD34-positive cells, showed an increase in LVEF after 6 and 12 months (Fig 7D and 7E). CD133-positive cells showed no improvement in any outcome (Fig 7A–7G). For ALDH-positive cells, no difference between cell therapy and control groups was observed in the outcomes (Fig 7A–7I). In response to treatment with BM-derived multipotent stromal cells (BM-MSc), the average difference in LVEF change was 6.1% (95% CI = 4.37–7.89, p<0.001) after 6 months (Fig 7D), but only 2.43% (95% CI = 7.75–12.50, p = 0.64; Fig 7E) at 12 months. For BM-derived "cardiopoietic cells", the LV end-diastolic dimensions were not changed after 6 months (-0.19, 95% CI = -0.47–0.1, p = 0.19; Fig 7F). However, after 12 months, LVED dimensions was significantly reduced (-0.39, 95% CI = -0.66 −-0.12, p = 0.005; Fig 7G). Moreover, exercise tolerance was improved and the incidence of SAEs was reduced (S4 and S5 Tables). For STRO-3-positive MPCs, *in vitro* cultured cells enriched for STRO-3 cell surface marker expression (one study with three arms of different

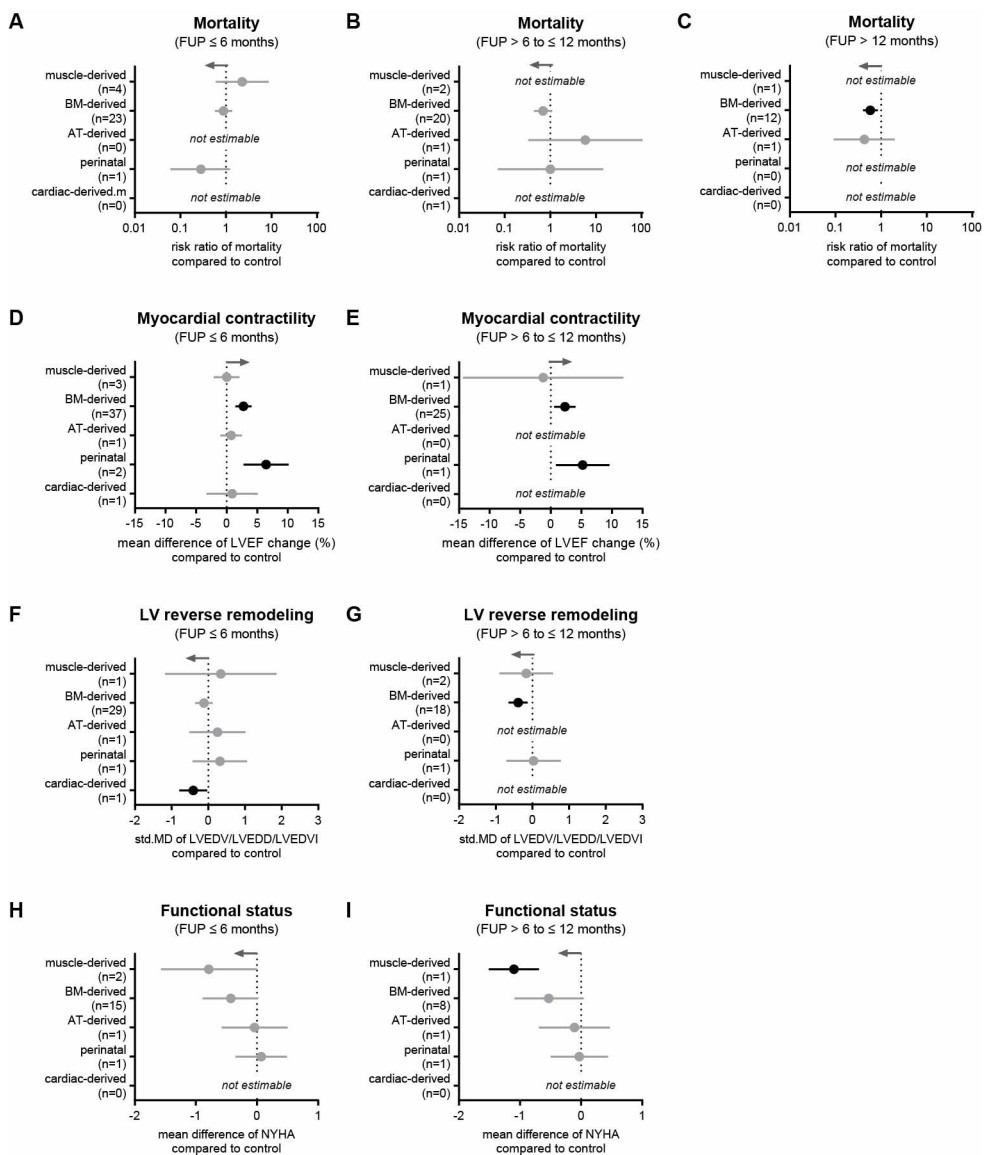

**Fig 6. Comparison of cells from different origins on clinical outcomes in HF patients.** (A-C) Mortality rates at 6, 12, and more than 12 months. (D, E) Change in LVEF at 6- and 12-month follow-ups. (F, G) Change in LV end-diastolic dimension at 6- and 12-month follow-ups. (H, I) NYHA functional status class at 6- and 12-month follow-ups. Gray arrows represent the trend in favor of cell therapy and n indicates the number of studies considered. Information on which studies were included in the analysis is shown in S3 Table.

cell doses) [47], there was no apparent improvement in heart function (Fig 7A–7G). Similar results were obtained for the lxmyelocel-T group (Fig 7A–7G) [29, 41].

## Effects of cells on different types of HF

Of the 33 RCTs using BM-derived cells, 24 studies were performed with patients with ischemic HF (1667 participants) and 6 with non-ischemic HF (498 participants); the remaining 3 studies included patients with both ischemic and non-ischemic HF (181 participants). In general, no difference in mortality was found between cell therapy with BM-derived cells and controls in any of the three subgroups (Fig 8A–8C). However, patients with ischemic or non-ischemic HF

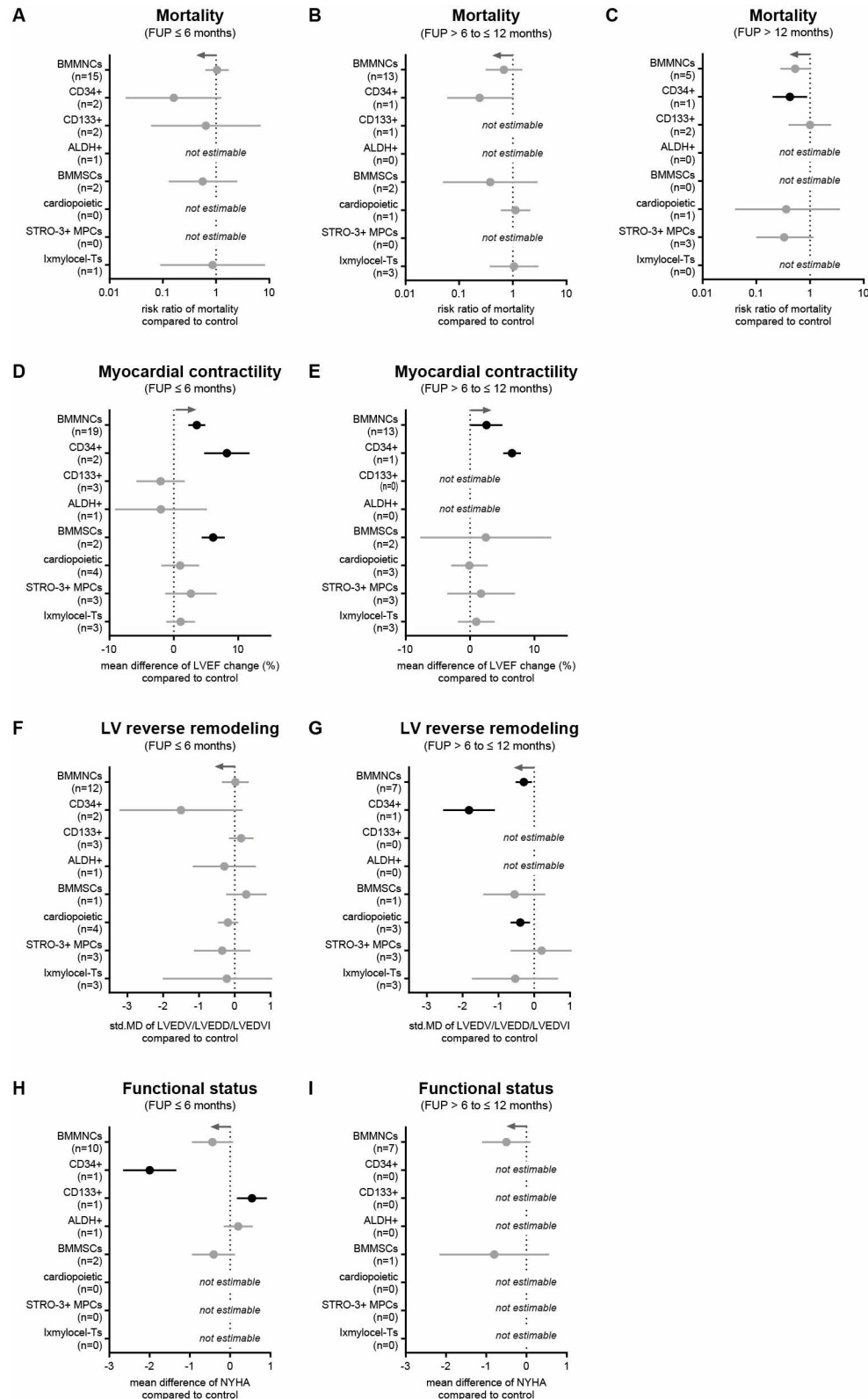

**Fig 7. Impact of different BM-derived cell types on clinical outcomes in HF patients.** (A-C) Mortality rates at 6, 12, and more than 12 months. (D, E) Change in LVEF at 6- and 12-month follow-ups. (F, G) Change in LV end-diastolic dimension at 6- and 12-month follow-ups. (H, I) NYHA functional status class at 6- and 12-month follow-ups. Gray

arrows represent the trend in favor of cell therapy and n indicates the number of studies considered. Information on which studies were included in the analysis is shown in S3 Table.

responded to cell therapy by the improvement of myocardial contractility after 6 months (Fig 8D). Ischemic HF patients who received cell therapy had an increase in LVEF of 2.49% (95% CI = 0.80−4.18, p = 0.004) compared to controls, while in non-ischemic HF patients LVEF change was 3.79%, 95% CI = 1.88−5.70, p = 0.0001; Fig 8D. After 12 months, myocardial contractility only significantly changed in non-ischemic HF patients after cell therapy (Fig 8E). LVED dimensions were not affected in all groups within the first 6 months after therapy (Fig 8F), but LV sizes became smaller in ischemic HF patients after 12 months (Fig 8G). NYHA class did not change in patients with ischemic or non-ischemic HF after 6 months (Fig 8H), but was lower after 12 months in non-ischemic HF patients (-0.53, 95% CI = -0.84–0.21, p = 0.001; Fig 8I). Further details on clinical outcomes are given in S4 and S5 Tables.

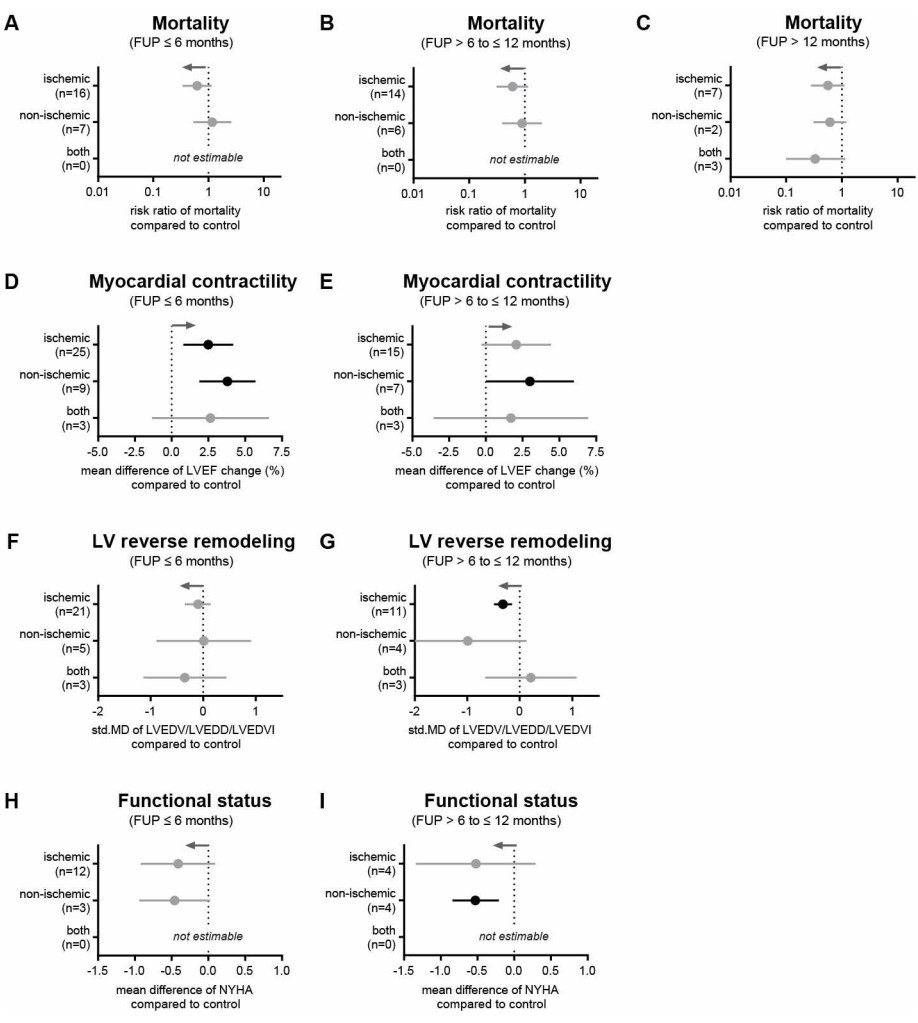

**Fig 8. Impact of cell therapy with BM-derived cells on different HF types.** (A-C) Mortality rates at 6, 12, and more than 12 months. (D, E) Change in LVEF at 6- and 12-month follow-ups. (F, G) Change in LV end-diastolic dimension at 6- and 12-month follow-ups. (H, I) NYHA functional status class at 6- and 12-month follow-ups. Gray arrows represent the trend in favor of cell therapy and n indicates the number of studies considered. Information on which studies were included in the analysis is shown in S3 Table.

## Impact of cell delivery route and revascularization on the outcome of intervention

Different cell delivery routes were used, including stand-alone surgical intramyocardial injection via left thoracotomy (IMI, 1 study and 1 extra arm; 69 participants), intracoronary injection (ICI, 7 studies and 1 arm; 646 participants), and transendocardial cell injection (TESI, 12 studies and 2 extra arms; 1104 participants). In addition, concomitant surgical revascularization by coronary artery bypass grafting (CABG) was performed in 11 trials with 527 participants. Overall, there was no inter-group difference in mortality rate for any of the cell delivery routes mentioned after 6 and 12 months (Fig 9A and 9B). During long-term FUP for more than 12 months, only TESI was associated with a lower RR of mortality (Fig 9C). At 6 months, HF patients who received cells by TESI and ICI had improved LVEF. Contractility was changed by 5.5% (0.24–10.76) 12 months after CABG+ICI therapy (Fig 9E). In addition, we found that none of the various cell delivery routes affected LVED dimensions within the first 6 months (Fig 9F), but it changed within 12 months after CABG +ICI and TESI therapy (Fig 9G). Only patients treated through ICI showed a difference in the NYHA score compared to controls (Fig 9H and 9I). Furthermore, as shown in Fig 10A–10C, patients who received surgical revascularization generally had lower mortality rates compared to patients who did not undergo surgical revascularization. In detail, 6 months after intervention, pooled death rates ranged between 1.40% and 4.00% for patients who had revascularization, while they ranged between 3.83% and 17.07% for patients without revascularization (Fig 10A). After 12 months and at long-term FUP, mortality rates increased to 6.90% or 20.21% in patients with revascularization, and to 21.31% or 34.74% in patients without revascularization, respectively (Fig 10B and 10C). In Fig 10D, bubbles representing the mean LVEF value at 6 months after intervention are evenly distributed along the x-axis but not on the y-axis. Bubbles representing the revascularization group are scattered around the line of 5% LVEF change, while bubbles representing patients without revascularization are mainly accumulated below 5% LVEF change. Fig 10E shows that the change in LVEF was significantly higher in patients who received revascularization compared to those who did not. In detail, revascularization combined with cell-based therapy resulted in a LVEF change from 3.7±9.9% to 9.8±8.4% (p<0.0001). In contrast, revascularization without cell-based therapy led to a LVEF change from 1.4±6.6% to 3.7±6.6% (p<0.001). Similar findings for the LVEF change were observed at 12 months after intervention. However, no long-term benefit of revascularization was observed for patients who did not receive cell transplantation (p = 0.747) (Fig 10F and 10G).

## Impact of cell dose on the outcome of intervention

First, we performed meta-regression analyses to investigate the correlation between cell doses and effect sizes. As shown in Fig 11A–11C, however, no correlation between the cell dose and the RR of mortality was determined. Similar results were found for the correlation between cell doses and LVEF change (Fig 11D and 11E). Then, the studies were stratified as follows: (i) less than 1 million cells (1 study, 63 participants), (ii) 1–10 million cells (22 studies and 2 arms, 1439 participants), (iii) 10–100 million cells (4 studies and 2 arms, 259 participants), and (iv) more than 100 million cells (3 studies and 3 arms, 585 participants). Fig 12A–12I show that no difference was observed between cell therapy and control groups in HF patients receiving less than 1 million cells, but the incidence of MACEs and SAEs after 6 months of FUP was lower (S5 Table). For patients receiving 1–10 million cells, no reduction in mortality rates was observed (Fig 12A–12C), but the LVEF change increased from 2.00% at 6 months to 3.28% at 12 months (Fig 12D and 12E). LV remodeling was not reversed after 6 months (Fig 12F), but

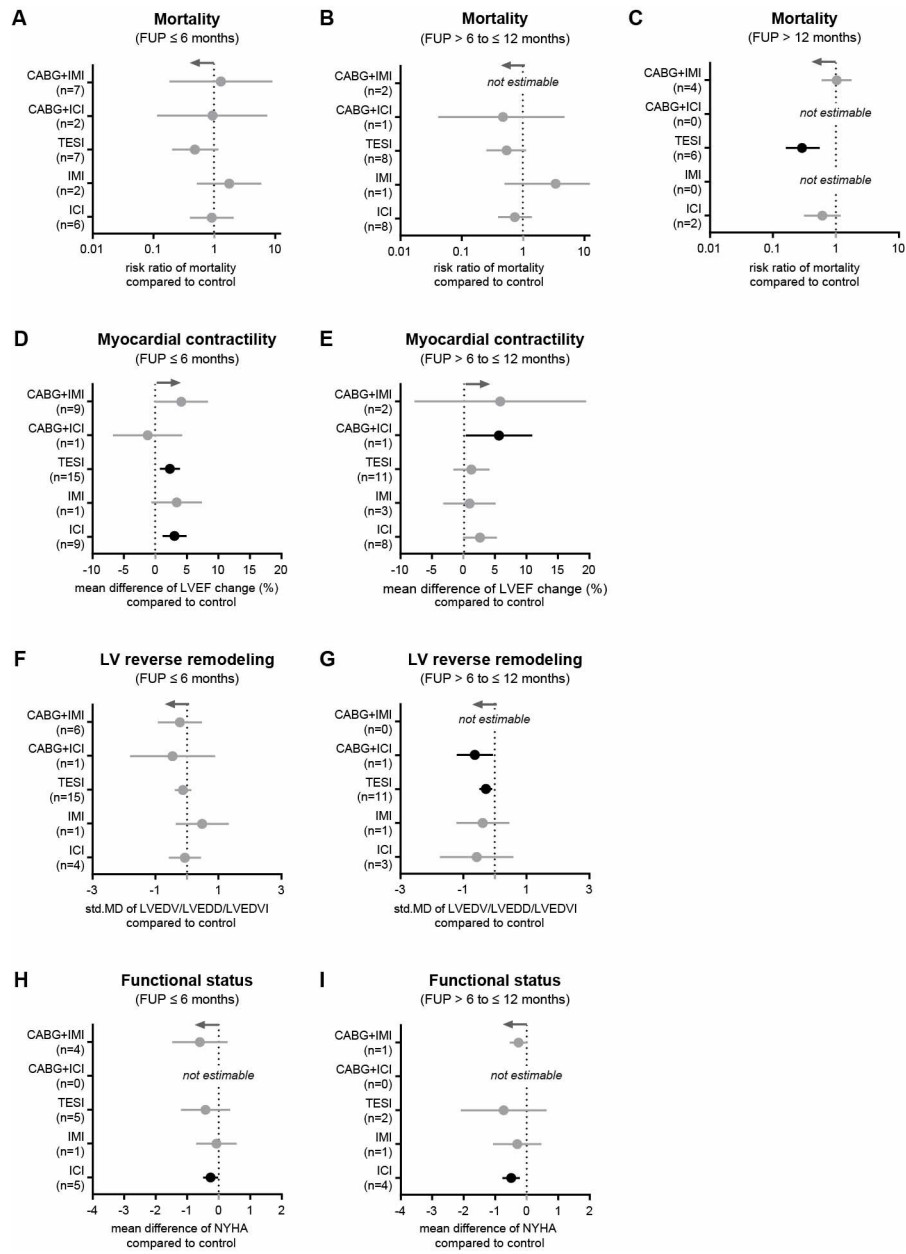

**Fig 9. Impact of different cell delivery routes on clinical outcomes in patients with HF.** (A-C) Mortality rates at 6, 12, and more than 12 months. (D, E) Change in LVEF at 6- and 12-month follow-ups. (F, G) Change in LV end-diastolic dimension at 6- and 12-month follow-ups. (H, I) NYHA functional status class at 6- and 12-month follow-ups. Gray arrows represent the trend in favor of cell therapy and n indicates the number of studies considered. Information on which studies were included in the analysis is shown in S3 Table.

after 12 months (Fig 12G). In contrast, patients who received 10−100 million cells had a LVEF improvement of 5.23% (Fig 12D) and ameliorated HF symptoms (Fig 12H) after 6 months, but no improvement in clinical outcomes was found after 12 months (Fig 12I). For patients receiving more than 100 million cells, a reduction in mortality rates was observed after 6 months and after more than 12 months (RR = 0.18, 95% CI = 0.03−0.97, p = 0.046 and RR = 0.41, 95% CI = 0.21−0.8, p = 0.01, respectively; Fig 12A and 12C). In addition, a reversal of LV remodeling was observed during FUP examinations (Fig 12F and 12G).

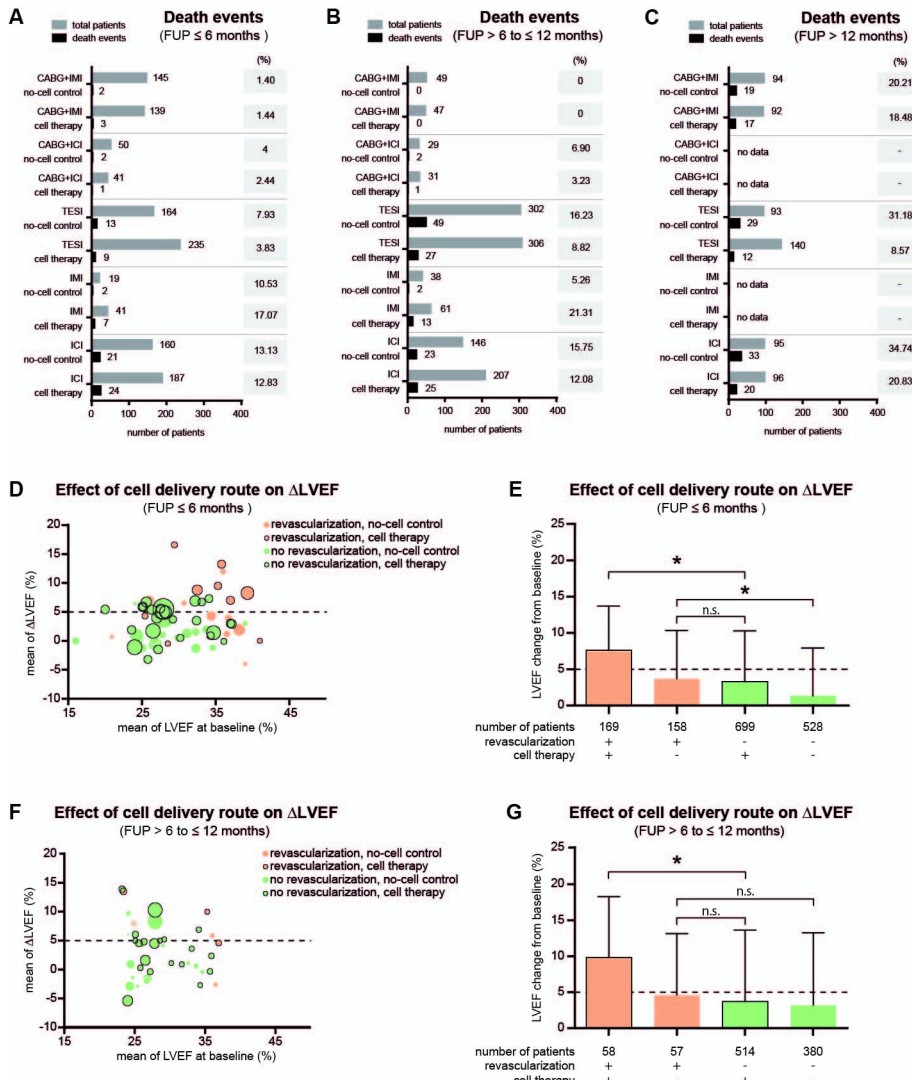

**Fig 10. Impact of different cell delivery routes on death events and LVEF change in HF patients.** (A-C) Incidence of pooled death events compared to total patient numbers at 6-, 12-, and more than 12-month follow-ups. Percentages of death are shown in the columns on the right side. (D-E) Impact of different delivery routes on LVEF change after 6-months follow-up. (F-G) Impact of different delivery routes on LVEF change after 12-months follow-up. In (E) and (G), asterisks indicate p-values of less than 0.001.

## Discussion

### Summary of the main findings

We meta-analyzed the results of 43 RCTs with a total of 2855 participants who underwent myocardial cell therapy for chronic ischemia and/or HF and found that (i) the average reported gain in LVEF is only 2,5%, indicating a lack of clinical significance, (ii) primary cells may yield slightly better results than *in vitro* cultured cells, and (iii) a meaningful benefit is only observed when revascularization (i.e. CABG) is done in conjunction with cell therapy. Overall, experimental design and procedural challenges significantly affect the robustness of therapeutic outcome variables, and trials with a low risk-of-bias (ROB) in selection and performance showed no improvement in contractility in response to cell therapy. Approximately

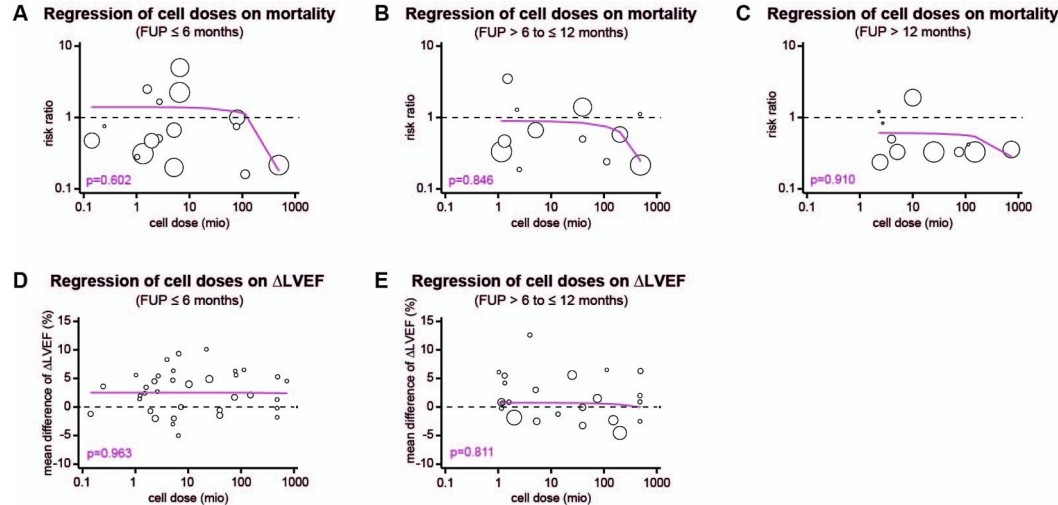

**Fig 11. Correlation of cell doses and primary outcomes.** (A-C) Meta-regression of cell doses on the RR of mortality rates after 6, 12, and more than 12 months after the intervention. (D-E) Meta-regression of cell dose on the increase of LVEF change from baseline after 6 and 12 months of follow-up.

twenty years ago, the highly publicized result of a number of experimental studies on myocardial cell therapy led to an unprecedented avalanche of clinical trials, in which somatic cells of presumed "regenerative" capacity were given to patients with acute myocardial infarction as well as those with chronic heart disease. Doubts regarding the biologic rationale were raised early on, but enthusiastic proponents outnumbered the more hesitant and the public eagerly awaited the soon-to-come cure of HF by myocardial regeneration. Even though the results of many clinical trials did not live up to expectations, the field continues to attract attention and consume significant resources.

A number of meta-analyses on cell therapy in HF patients have been carried out before. It is therefore understandable to question the need for another study on this subject, especially when there is already an authoritative study by Fisher et al., which is registered in the Cochrane library and updated every two years [68, 69]. The differences between Fisher's analysis and ours are summarized in Table 2. Rather than concentrating on periprocedural adverse events, we examined efficacy outcomes of LVEF change from baseline, which should reflect the regenerative effect of cell therapy in HF. Our more detailed subgroup analyses based on cell type etc. better reflect the heterogeneity of cardiac cell therapy approaches, albeit at the cost of biomathematical power.

## Improvement of LVEF as the primary efficacy outcome

Clinically, a change in LVEF greater than 5% is one of the standards for defining responders and non-responders to therapeutic interventions and it is an important predictor of prognosis in patients with HF [31, 70, 71]. Nevertheless, the use of LVEF as the primary efficacy outcome has its limitations. First, the LVEF value differs among different imaging techniques and is operator-dependent [72]. Second, Mitral regurgitation may lead to an overestimation of cardiac function due to retrograde emptying of the left ventricle, and eccentric hypertrophy in HF patients can lead to dysfunction of the mitral valve [73]. Nevertheless, patients with progressive mitral regurgitation but without surgical indications, were enrolled in many of the trials. We therefore used the LVEF change-from-baseline instead of the mean LVEF mean as an endpoint, because baseline fluctuations are minimized.

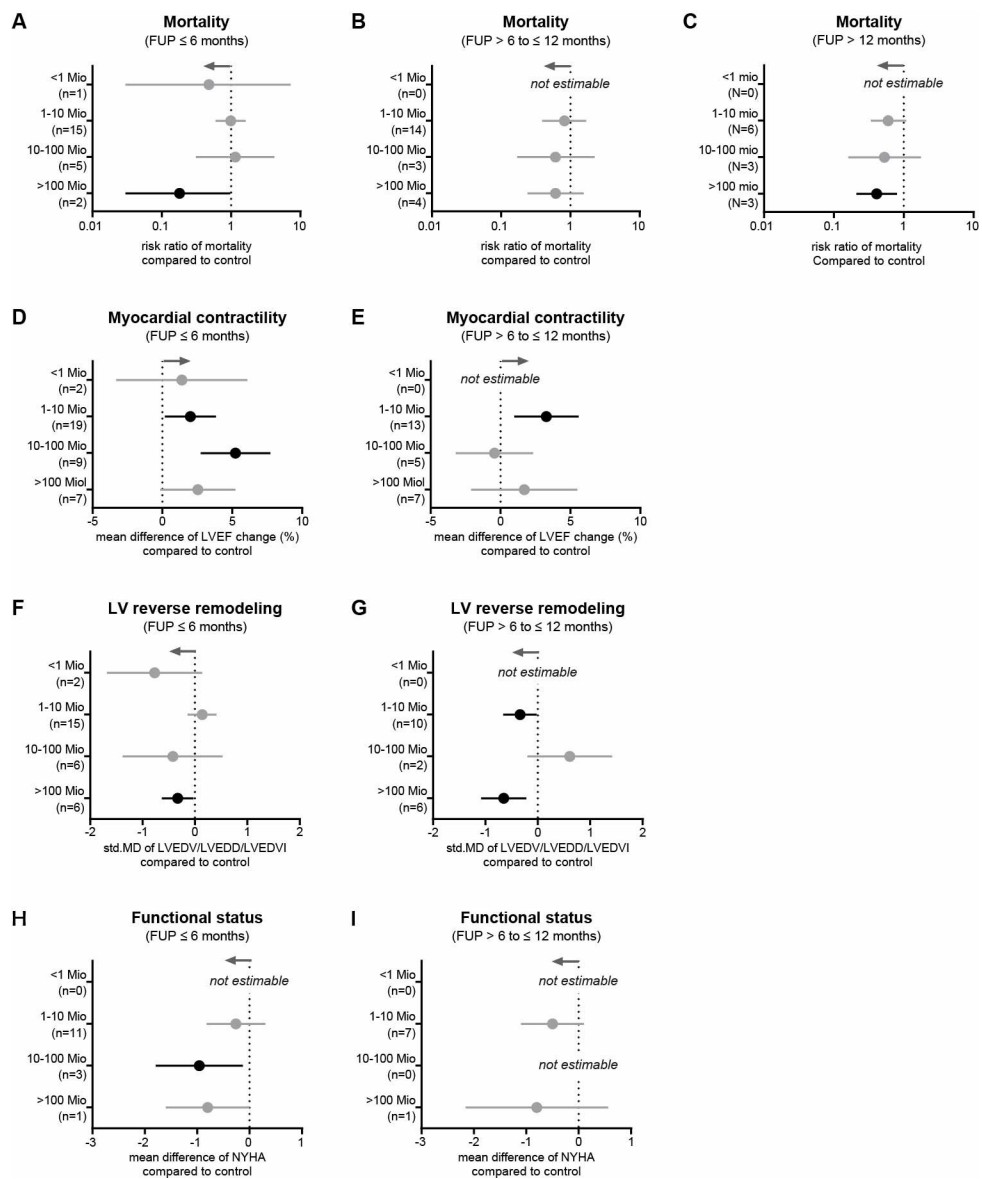

**Fig 12. Impact of different cell doses on clinical outcomes in HF patients.** Mortality rates at 6, 12, and more than12 months (D, E) Change in LVEF at 6- and 12-month follow-ups. (F, G) Change in LV end-diastolic dimension at 6- and 12-month follow-ups. (H, I) NYHA functional status class at 6- and 12-month follow-ups. Gray arrows represent the trend in favor of cell therapy and n indicates the number of studies considered. Information on which studies were included in the analysis is shown in S3 Table.

The overall increase in LVEF after the intervention was only 2.7% after 6 months of FUP and 2.5% at 12 months, which is in agreement with earlier meta-analyses [9, 74–77]. In experiments on small animals such as mice and rats, an increase in LVEF of around 12% in response to cell therapy has been reported, while in large animals such as sheep and pigs, a LVEF increase of 5–7% has been described [7, 8, 78, 79]. This difference has been attributed to the species-dependent size/dose mismatch, and the heterogeneity of clinical baseline situation (i.e. infarct size, interval) compared to an experimental model as well as the presence of HF-related chronic diseases such as diabetes and hyperlipidemia may also affect cell engraftment and their regenerative potential [80, 81].

**Table 2. Comparison between the analysis by Fisher et al. and our study.**

| | Fisher 2016 [69] | Our study |
|---|---|---|
| **Enrolled RCTs** | 38 | 43 |
| **Participants** | Anyone with a clinical diagnosis of IHD or CHF, but only ischemic HF | Ischemic or non-ischemic systolic decompensated HF with an LVEF of less than 50% and NYHA classes II-IV |
| **Primary outcomes** | Mortality | All-cause mortality |
| | Periprocedural adverse events | LVEF change from baseline |
| **Sensitivity analysis** | Exclusion of high/unclear ROB | Inclusion of results with both low and high/unclear risks of selection, performance and attrition biases on LVEF change |
| | Inclusion of results with a low risk of selection, performance and attrition bias | |
| **Subgroup analysis based on cell type** | Mononuclear cells | Muscle-derived cells |
| | Circulating progenitor cells | BM-derived cells |
| | Hematopoietic progenitor cells | |
| | Mesenchymal stromal cells | AT-derived cells |
| | | Perinatal cells |
| | | BMMNCs |
| | | CD34-positive cells |
| | | CD133-positive cells |
| | | ALDH positive cells |
| | | BMMSCs |
| | | STRO-3-positive cells |
| | | Cardiopoietic cells |
| | | Lxmyelocel-T |
| **Subgroup analysis based on cell dose** | ≤ 10 million cells | ≤ 1 million cells |
| | > 10 to ≤ 100 million cells | > 1 to ≤ 10 million cells |
| | > 100 million cells | > 10 to ≤ 100 million cells |
| | | > 100 million cells |

Cell therapy must not be an end in itself but a tool for improving symptoms and prognosis of patients with heart failure. As such, it competes with an ever-growing number of established and novel therapeutic interventions. The portfolio of interventions/medications for heart failure is too large to list here but one example is the heart rate reduction by Ivabradine, which inhibits the sodium-potassium inward current (If) that controls spontaneous diastolic depolarization in the sinoatrial node. As shown in the echocardiography substudy of the SHIFT trial, Ivabradine given on top of the standard heart failure medication reversed cardiac remodeling (evidenced by lower LVESDI, higher LVEF and fewer cardiovascular events) to an extent that is very similar to or greater than that seen after cell therapy.

Cell therapy must not be an end in itself but a tool for improving symptoms and prognosis of patients with heart failure. As such, it competes with an ever-growing number of established and novel therapeutic interventions. The portfolio of interventions/medications for heart failure is too large to list here but one example is the heart rate reduction by Ivabradine, which inhibits the sodium-potassium inward current (If) that controls spontaneous diastolic depolarization in the sinoatrial node. As shown in the echocardiography substudy of the SHIFT trial, Ivabradine given on top of the standard heart failure medication reversed cardiac remodeling (evidenced by lower LVESDI, higher LVEF and fewer cardiovascular events) to an extent that is very similar to or greater than that seen after cell therapy [82].

## Bias in outcome reporting

Especially in a field as controversial as cardiac cell therapy, a strict process is necessary to generate reproducible and convincing results. Despite modern biomathematical tools, a meta-analysis is only as valid as the included trials are. Of the 43 RCTs, 23 publications did not report the details of either generating a random sequence or allocating patients to each trial arm. Schulz et al. estimated that unclear report during randomization is associated with up to 30% exaggeration of the effect estimates over the true effect [83]. We found that trials with unclear randomization reported an over 3% higher LVEF increase compared to controls than trials done with adequate randomization. Proper masking is similarly important in RCTs. Breaking blindness causes performance bias by influencing the investigators' care levels, and by impacting patient behavior [84–86]. Here, 14 trials did not perform well in masking, and in 2 trials details are missing. Most of the trials were designed as open-label trials, without cell harvest and sham injection in control patients. For instance, in one trial only patients in the treatment group received intracoronary delivery of allogeneic UCMSCs while control patients received conventional medical treatment [61]. Again, while the LVEF increase in trials with a low risk of performance bias was negligible, the average effect size in trials with high or unclear risk was approximately 5% higher.

The DAMASCENE trial by Nowbar et al. was the first to systematically study the impact of reporting quality on effect size in cardiac cell therapy.[11] They found that the number of discrepancies in trial reports positively correlates with the effect size (LVEF). Incompatible data and contradictory descriptions are often accidentally generated during manuscript drafting and revising, and not every discrepancy indicates data manipulation. However, they often arise from the tremendous pressure on authors to meet "expectations", and to do so quickly. Eleven trials included in this meta-analysis were also evaluated in DAMASCENE. Incompatible data were found among different reports from the same trial [4, 53, 54], but were mostly due to different result presentations in different phases of the clinical study. Incompatible data in plots and tables were also detected [57, 60]. The DAMASCENE paper has been challenged and compared with an ecological fallacy, erroneously constructing a cause-effect relationship between independent observations [87]. This may be arguable from a purely epistemologic point-of-view, but, given the accumulation of irregularities in this particular field, DAMASCENE still provides valuable insight in the problems surrounding cardiac cell therapy.

## Factors affecting cell therapy outcomes

**Cell origin.** Up to this day, most of the stem cells used in clinical trials for HF treatment were autologous BM-derived cells, mainly from elderly HF patients, which are known to have impaired functional capacity. Perinatal mesenchymal stromal cells have been used in some clinical trials. They may bear the benefit of not being associated with age-related issues such as DNA damage and telomere dysfunction [88–90]. Therefore, they may be more resistant to reactive oxygen species and could retain greater therapeutic effectiveness under adverse conditions during transplantation [91]. Improvements in heart function have been attributed to paracrine cardioprotective and pro-angiogenic effects rather than the re-generation of contractile cells [92]. One RCT used heart-derived cells but showed no difference in primary outcomes compared to the control group [32], similar to the initial trials using skeletal myoblasts. Taken together, the tissue origin of the cell product does not influence clinical efficacy.

**Cell phenotype.** Beneficial effects on LV function were reported in a number of trials using primary BM-derived cells, while transplantation of enriched CD133-positive BM cells, a subset of BMMNCs, had no effect. For cultured BM-derived cells, BM-MSCs were shown to improve contractility and reverse ventricular remodeling, while BM-derived cardiopoietic

cells, STRO-3-positive MPCs, and Ixmyelocel-T did not exhibit similar effects. In two clinical studies, both conducted by Bartunek and colleagues [63, 64], patients received approximately 600 million "cardiopoietic" cells and showed an almost 6% increased LVEF in the C-CURE study [63], but a decreased LVEF in the CHART-1 trial [64], and LVEDV behaved similarly. The only methodologic difference between the two trials was group sizes (47 participants in the C-CURE study and 348 participants in the CHART-1 trial), emphasizing that a smaller sample size increases the likelihood of chance findings. In the study by Perin [47], dose-escalated STRO-3-positive MPCs did not lead to consistent dose-dependent effects. The greatest LVEF improvement occurred at a dose of 25 million cells, while 150 million MPCs had negative effects on LVEF but performed best in reducing LV sizes. Again, chance is the best explanation for these conflicting findings.

In general, maintaining progenitor cell quality during *in vitro* culture is still a major challenge [93]. This could be one reason why consistent results have been difficult to obtain. The properties of proliferating and/or immature cells can be affected by medium components such as basal medium and serum [94, 95]. For example, fetal bovine serum has been reported to cause cell cycle termination and suppression of genes related to proliferation [96]. Therefore, *in vitro* cultured cells may be particularly ill-suited for cell therapy [97, 98].

**Cell delivery route and timing.** Regardless of whether cells are delivered via IMI, ICI, or TESI, cell loss occurs gradually over time after cell injection. Cell tracing experiments in both pigs and rats showed that after 24 hours of transplantation, cell retention rates were less than 10% after ICI and approximately 20% after IMI. After one month, cell retention rates for both methods are below 1% [99–101]. Therefore, in addition to analyzing the differences between various cell delivery methods, we also looked at the impact of concomitant revascularization which, performed either by CABG or percutaneous coronary intervention, yields a significant survival benefit in patients with chronic HF [102, 103]. We found that patients who received CABG surgery plus cell transplantation had neither reduced mortality nor improved heart function compared to patients who received CABG alone. Only one study conducted by Hu and colleagues [31] suggested that the use of CABG surgery in combination with cell therapy is superior to CABG surgery alone.

We were also interested to see whether the timing of intraoperative cell transplantation in relation to revascularization has an impact on the change in LVEF (Table 3). During CABG surgery on cardiopulmonary bypass and cardioplegic arrest, cells were injected after the coronary anastomoses were completed and right before the aortic cross-clamp was opened, in the non-perfused myocardium. Overall, patients receiving this type of therapy showed a mean LVEF increase of less than 5%, regardless of the duration of FUP [35, 42, 46]. In the study of Hu et al. [31], cells were injected through the bypass grafts after the distal anastomosis was completed, resulting in an LVEF increase by 5.5% (0.24–10.76) compared to controls. In the study by Trifunovic et al. [56], cells were injected into beating hearts after weaning from cardiopulmonary bypass. Here, LVEF improved by 8.3% [5–11.6] at 6 months and by 12.6% (8.96–16.24) at 12 months. In two other studies, cells were injected into the beating heart during off-pump CABG surgery, and the gain in LVEF was significantly higher than on CABG-only patients (6.35% (4.11–8.59) [58] and 9.35% (4.76–-13.94) [60]). At first glance, these results are encouraging. On the other hand, in several trials combining CABG and cell therapy no relevant change in LV function was observed at all [37, 38, 42, 46, 55]. CABG and cell therapy is a complex intervention with high inter-individual variability. CABG alone should increase LF function when ischemic viable or hibernating myocardium is reperfused, but myocardial viability in the target area is not always determined before surgery. Also, cohort size in those trials tends to be small, so that the results should be viewed with caution.

**Table 3. Comparison of LVEF results based on the time point of cell delivery in patients who received BMMNCs during surgical revascularization.**

| Study | Timing of cell transfer | LVEF change from baseline (mean difference) | |
|---|---|---|---|
| | | FUP ≤ 6 months | FUP > 6 to ≤ 12 months |
| | | mean (95% CI) | mean (95% CI) |
| Hu 2011 [31] | Before completion of distal anastomosis of coronary arteries | - | 5.50% (0.24−10.76) |
| Ang 2008 [46] | Before unclamping | IMI: 3.60% (-1.28−8.48) | - |
| | | ICI: -1.20% (-6.75−4.35] | |
| Maureira 2012 [35] | Before unclamping | 4.00% (-3.63−11.63) | - |
| Patila 2014 [42] | Before unclamping | - | -1.24% (-5.25−2.77) |
| Trifunovic 2015 [56] | After successful weaning from CPB | 8.30% (5.00−11.60) | 12.60% (8.96−16.24) |
| Wang 2015 [58] | Off-pump CABG | 6.35% (4.11−8.59) | - |
| Zhao 2008 [60] | Off-pump CABG: 22/36 patients | 9.35% (4.76−13.94) | - |
| | On-pump CABG: 14/36 patients | | |

BMMNCs were injected intramyocardially into the heart unless otherwise mentioned.

**Cell dose.** The retention rate of transplanted cells in the heart is low and depends on the mode of delivery as well as on the cell product [104]. Given the unclear mechanism-of-action, it is impossible to predict the biological response to a given cell dose. In large animals, an increase in LVEF of approximately 6% has been reported at a dose of less than 10 million cells and of approximately 8% for a dose of equal to or greater than 10 million cells [8]. In an attempt to account for the differences in cell preparation, especially unfractionated BMMNC, we estimated the number of "active" progenitors in each cell product based on the percentage of CD34-positive cells. We used CD34 because (i) CD34 is widely regarded as a biomarker for juvenile cells and used in cell sorting [18, 19], (ii) many of the enrolled trials reported the amount of CD34-positive cells, and (iii) dose-response effects of CD34-positive cells and the increase in LVEF have been demonstrated by several exploratory analyses [4, 105, 106]. Our meta-regression analysis showed no positive correlation between cell dose and LVEF gain. On the contrary, the average LVEF change decreased at cell doses greater than 100 million compared to 10–100 million. Preclinical dose-escalation studies did not show a consistent dose-response, either [8, 107], and only in one meta-analysis of clinical cell therapy in acute myocardial infarction evidence of a positive correlation between cell dose and LV function was found [108]. Given that a dose-response relationship is one of the hallmarks of a functioning drug, these inconsistencies underscore the overall failure of cardiac cell therapy.

**Heterogenous patient response.** This meta-analysis is based on the overall-results of the included trials. In a number of reports, however, attempts were made to identify and characterize patient subpopulations that benefit from cell therapy more than the cohort average suggests. One such phenomenon is that the benefit of the therapeutic intervention inversely correlates with baseline LV function. Obviously, when LV function is only moderately impaired, improvements are more difficult to detect than in patients with decidedly poor baseline contractility. Also, the myocardium may be more amenable to repair in patients with recent onset of disease, compared to those with long-standing heart failure and severely scarred/fibrotic myocardium. A machine-learing based post-hoc analysis of patients treated with CD133+ BMC in the PERFECT trial revealed that mutations of the SH2B adapter protein

3 (SH2B3), also known as lymphocyte adapter protein (LNK), in hematopietic stem cells predict the outcome of cardiac cell therapy [109]. Hence, it may be that cell therapy has a reproducible effect in selected patients who share the above-mentioned and other characteristics. In the real-world-scenario, however, it would be laborious and impractical to identify such "responder" patients prior to cell therapy.

Similarly, autologous cell products have inevitably some inter-individual variability with respect to effector cell dose, number of contaminating cell types, as well as age- and disease-related changes in effector cell biology. Allogenic cell products are more homogenous, at least those derived of a single batch. A more stringent trial design using truly uniform cell products may reduce these confounders but, again, at the cost of more complex cell preparation and quality control measures.

## Limitations

This meta-analysis has several limitations. First, missing data was requested from corresponding authors, but only one author responded. As a result, some data had to be extracted from high-resolution graphs, which has proven to be feasible and efficient [110, 111]. In addition, data on change from baseline had to be calculated for LVEF, LVEDV, and LVEDVI where these specific data were missing. Second, because of the stratification into subgroups, sometimes only one or two studies were available for analysis, with inevitable consequences for the power of the analysis.

## Conclusion

After two decades of clinical trials on cell therapy for heart disease, there is no overall trend that would indicate any meaningful therapeutic efficacy. Neither the characteristics of the cell products nor any patient- or procedure-related variables indicate that a particular approach may at least be promising. The lack of clinical evidence together with conflicting preclinical data and limited biologic plausibility regarding the plasticity of non-pluripotent progenitor cells indicate that cardiac cell therapy in its present form has no therapeutic effect. Further clinical applications should not be performed before fundamentally novel and plausible strategies with a sound scientific basis have been developed.

## Supporting information

**S1 Fig. Evaluation of publication bias in primary outcomes.**
(TIF)

**S2 Fig. Forest plots.** (A) overall LVEF change from baseline at 6-month follow-up, (B) overall LVEF change at 12-month follow-up, (C) overall mortality at 12-month follow-up (D) overall mortality at 12-month follow-up and (E) overall mortality in follow-ups more than 12 months. (TIF)

**S1 Table. Inclusion and exclusion criteria of studies for meta-analysis.** Patients undergoing LVAD implantation were excluded because most deaths were due to LVAD dysfunctions, pump thrombus, multi-system organ failure, and sepsis that are likely to be irrelevant for cell therapy applications.
(DOCX)

**S2 Table. Overview of primary and secondary outcomes.**
(DOCX)

**S3 Table. Overview of studies included in each analysis.** Specific study arms are given in brackets.
(DOCX)

**S4 Table. Summary of efficacy outcomes after cell therapy.** The first number represents the number of comparisons, the second number indicates the number of participants, the third number represents the effect size [95% confidence interval] and the fourth number indicates the value of $I^2$ index.
(DOCX)

**S5 Table. Summary of safety outcomes after cell therapy.** The first number represents the number of comparisons, the second number indicates the number of participants, the third number represents the effect size [95% confidence interval] and the fourthl number indicates the value of $I^2$ index.
(DOCX)

**S1 Text. PRIMSA checklist.**
(DOC)

**S2 Text. Electronic search strategy used in the meta-analysis.** The numbers in brackets indicate the number of studies.
(DOCX)

## Acknowledgments

We thank Robert Röhle from the Coordination Center for Clinical Studies and the Institute of Biometry and Clinical Epidemiology of the Charité–Universitätsmedizin Berlin for statistical support.

## Author Contributions

**Conceptualization:** Zhiyi Xu, Christof Stamm.

**Data curation:** Zhiyi Xu, Sebastian Neuber, Timo Nazari-Shafti, Zihou Liu, Fengquan Dong.

**Formal analysis:** Sebastian Neuber, Zihou Liu, Fengquan Dong, Christof Stamm.

**Investigation:** Zhiyi Xu, Zihou Liu.

**Methodology:** Zhiyi Xu, Fengquan Dong, Christof Stamm.

**Supervision:** Timo Nazari-Shafti, Christof Stamm.

**Validation:** Timo Nazari-Shafti, Fengquan Dong, Christof Stamm.

**Writing – original draft:** Zhiyi Xu, Sebastian Neuber, Timo Nazari-Shafti, Zihou Liu, Fengquan Dong.

**Writing – review & editing:** Christof Stamm.

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
