## [Decision Letter · Decision Letter 0]

14 Jun 2021

PONE-D-21-12505

Impact of procedural variability and study design quality on the efficacy of cell-based therapies for heart failure - a meta-analysis

PLOS ONE

Dear Dr. Stamm,

Thank you for submitting your manuscript to PLOS ONE. After careful consideration, we feel that it has merit but does not fully meet PLOS ONE’s publication criteria as it currently stands. Therefore, we invite you to submit a revised version of the manuscript that addresses the points raised during the review process.

All issues raised by expert reviewers are required.

We look forward to receiving your revised manuscript.

Kind regards,

Vincenzo Lionetti, M.D., PhD

Academic Editor

PLOS ONE

Journal Requirements:

2. Thank you for stating the following in the Financial Disclosure Statement Section of your manuscript:

"We thank the China Scholarship Council for financial support to XZ, FD and ZL. Funders had no role in

study design, data collection and analysis, decision to publish, or preparation of the manuscript."

"The authors received no specific funding for this work."

4. We note that this manuscript is a systematic review or meta-analysis; our author guidelines therefore require that you use PRISMA guidance to help improve reporting quality of this type of study. Please upload copies of the completed PRISMA checklist as Supporting Information with a file name “PRISMA checklist”.

Reviewers' comments:

Reviewer's Responses to Questions

**Comments to the Author**

1. Is the manuscript technically sound, and do the data support the conclusions?

Reviewer #1: Yes

Reviewer #2: Yes

2. Has the statistical analysis been performed appropriately and rigorously? 

Reviewer #1: Yes

Reviewer #2: Yes

3. Have the authors made all data underlying the findings in their manuscript fully available?

Reviewer #1: Yes

Reviewer #2: Yes

4. Is the manuscript presented in an intelligible fashion and written in standard English?

Reviewer #1: Yes

Reviewer #2: Yes

5. Review Comments to the Author

Reviewer #1: In the current study, the authors have conducted an analysis of the studies focusing on the use of different cell types for heart failure (HF). It is an elaborate report on cell therapy with a focus on the types of cells including both primary as well as cultured cells, sourced from bone marrow, skeletal muscle, perinatal, adipose tissue and cardiac derived cells. The authors also assessed the effect of cell therapy on different types of heart failure. Additionally, the authors included the assessment of the mode of cell delivery and its influence on the study findings. Finally, they have looked how the changes in the cell numbers during transplantation affect the eventual outcome of the study. The study follows the PRIMSA guidelines for a meta-analysis.

Though the authors have highlighted major findings of the studies involving cell therapy for HF and covered wide range of topics. I believe it would benefit the readers to include a section in the manuscript on future directives summarizing some suggestions on how to have more defined conditions to improve the study design and have more consistent trial procedures; in order to rule out the fact that the non-significant changes seen post transplantation exist due to variations in preparation of cells in the lab or might be the result of inefficiency of the cells by themselves.

Reviewer #2: The authors systematically reviewed 43 RCTs of cell therapy for HF that enrolled a total of 2855 patients, with the majority (33 studies, 2346 participants) evaluating bone marrow-derived cells. They found that, overall, cardiac cell therapy increased LVEF by 2.7% after 6 months and by 2.5% after 12 months, did not have significant effect on all-cause mortality within 6 and 12 months (whereas a significant reduction was found for RCTs lasting >12 months), and did not affect the rate of MACE and SAE. The effect on secondary, less impactful outcomes (e.g. biomarkers) was variable. The results of RCTs with primary cells and in which cardiac cell therapy was combined with surgical revascularization were better. Of note, many RCTs suffered from significant risk of bias, and no significant improvement in LVEF was found in the RCTs with low risk of bias.

The article is well-written and methodologically sound. The originality is limited , because other meta-analysis were published on the topic and one is regularly updated in the Cochrane library. However, the authors acknowledge this limitation and overtly discuss the similarities and differences with the Cocharne meta-analysis.

I recommend the publication of the paper. However, the authors may want to acknowledge that there may have been patient subsets, not taken into account by this study-level meta-analysis, that did benefit from cardiac cell therapy. In this regard, it may be interesting – and provoking – comment the results of this meta-analyses with the echo study of the SHIFT trial: European Heart Journal (2011) 32, 2507–2515

6. PLOS authors have the option to publish the peer review history of their article (what does this mean?). If published, this will include your full peer review and any attached files.

Reviewer #1: **Yes: **Sanjiv Dhingra

Reviewer #2: No

---

## [Author Response · Author response to Decision Letter 0]

11 Nov 2021

Please find the details of the revision in the file named "Response to Reviewers".

---

## [Decision Letter · Decision Letter 1]

3 Dec 2021

Impact of procedural variability and study design quality on the efficacy of cell-based therapies for heart failure - a meta-analysis

PONE-D-21-12505R1

Dear Dr. Stamm,

We’re pleased to inform you that your manuscript has been judged scientifically suitable for publication and will be formally accepted for publication once it meets all outstanding technical requirements.

Kind regards,

Vincenzo Lionetti, M.D., PhD

Academic Editor

PLOS ONE

Additional Editor Comments (optional):

Reviewers' comments:

Reviewer's Responses to Questions

**Comments to the Author**

1. If the authors have adequately addressed your comments raised in a previous round of review and you feel that this manuscript is now acceptable for publication, you may indicate that here to bypass the “Comments to the Author” section, enter your conflict of interest statement in the “Confidential to Editor” section, and submit your "Accept" recommendation.

Reviewer #1: All comments have been addressed

Reviewer #2: All comments have been addressed

2. Is the manuscript technically sound, and do the data support the conclusions?

Reviewer #1: Yes

Reviewer #2: Yes

3. Has the statistical analysis been performed appropriately and rigorously? 

Reviewer #1: N/A

Reviewer #2: Yes

4. Have the authors made all data underlying the findings in their manuscript fully available?

Reviewer #1: Yes

Reviewer #2: Yes

5. Is the manuscript presented in an intelligible fashion and written in standard English?

Reviewer #1: Yes

Reviewer #2: Yes

6. Review Comments to the Author

Reviewer #1: Authors have addressed all the concerns, the manuscript looks much better now, no further comments..

Reviewer #2: The authors addressed my remarks. I recommend the publication of this article, which provides a rigorous synthesis of the evidence on cell-based therapies for HF

7. PLOS authors have the option to publish the peer review history of their article (what does this mean?). If published, this will include your full peer review and any attached files.

Reviewer #1: No

Reviewer #2: **Yes: **Pietro Ameri

---

## [Editor Report · Acceptance letter]

27 Dec 2021

PONE-D-21-12505R1 

Impact of procedural variability and study design quality on the efficacy of cell-based therapies for heart failure - a meta-analysis 

Dear Dr. Stamm:

I'm pleased to inform you that your manuscript has been deemed suitable for publication in PLOS ONE. Congratulations! Your manuscript is now with our production department. 

Kind regards, 

on behalf of

Prof. Vincenzo Lionetti 

Academic Editor

PLOS ONE